# MAMUT: A Novel Framework for Modifying Mathematical Formulas for the Generation of Specialized Datasets for Language Model Training

**Jonathan Drechsel**                                    *jonathan.drechsel@uni-passau.de*
*Faculty of Computer Science and Mathematics, University of Passau, Germany*

**Anja Reusch**                                          *anja@campus.technion.ac.il*
*Taub Faculty for Computer Science, Technion - Israel Institute of Technology, Israel*

**Steffen Herbold**                                      *steffen.herbold@uni-passau.de*
*Faculty of Computer Science and Mathematics, University of Passau, Germany*

**Reviewed on OpenReview:** `https://openreview.net/forum?id=khODmRpQEx`

## Abstract

Mathematical formulas are a fundamental and widely used component in various scientific fields, serving as a universal language for expressing complex concepts and relationships. While state-of-the-art transformer models excel in processing and understanding natural language, they encounter challenges with mathematical notation, which involves a complex structure and diverse representations. This study focuses on the development of specialized training datasets to enhance the encoding of mathematical content. We introduce Math Mutator (MAMUT), a framework capable of generating equivalent and falsified versions of a given mathematical formula in LaTeX notation, effectively capturing the mathematical variety in notation of the same concept. Based on MAMUT, we have generated four large mathematical datasets containing diverse notation. Experiments show that models trained on these datasets exhibit new SoTA performance on mathematical retrieval tasks. We publish our code, generated datasets, and pretrained mathematical models: https://github.com/aieng-lab/math-mutator.

## 1 Introduction

Mathematical formulas are a fundamental and widely used component in various scientific fields, serving as a universal language for expressing complex concepts and relationships. Their context-dependent symbols, nested operations, and diverse notations pose distinct challenges for machine learning models due to their symbolic and structural differences from natural language (Zanibbi et al., 2020; Peng et al., 2021).

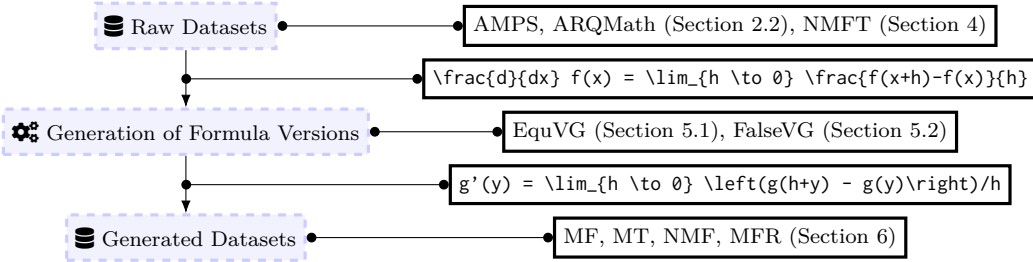

Figure 1: MAMUT: Modifying formulas to generate large and diverse mathematical datasets.

| Dataset | Description | Example(s) |
|---------|-------------|------------|
| Mathematical Formulas (MF) | Mathematical formulas with high variance | $x \cdot x^N = x^{1+N}$ 
 $(a-b)/(b*a) = -1/a + \frac{1}{b}$ |
| Mathematical Texts (MT) | Texts combining natural language and mathematical formulas | Identify $\sum_{n=0}^{\infty}(y_n - L)$ where $y_{n+1} = (1+y_n)^{\frac{1}{3}}$ and $L^3 = L+1$. Let $y > 2$ and let $f(y) = (1+y)^{\frac{1}{3}}$. Let $f^n(y)$ be the $n$ th iterate of $f(y)$. Let $L$ be ... |
| Named Mathematical Formulas (NMF) | High variance formulas of famous named identities | Name: Pythagorean Thm., Formula: $c^2 = b^2 + a^2$ 
 Name: Binomial Formula, Formula: $(\alpha + z)^2 = z^2 + \alpha^2 + 2 \cdot \alpha \cdot z$ |
| Mathematical Formula Retrieval (MFR) | Pairs of formulas with labels indicating identical or different mathematical concepts | Formula 1: $1 \cdot 2 \cdot 3 \cdot \ldots \cdot n = n!$, Formula 2: $m! := \prod_{k=1}^{m} k$, Label: Equivalent 
 Formula 1: $a^2 + b^2 = c^2$, Formula 2: $a^2 + 2^b = c^2$ Label: Not Equivalent |

Table 1: Overview of generated datasets. The examples are shown as rendered LaTeX.

Despite the success of transformer-based language models (Vaswani et al., 2017) in natural language tasks, they encounter challenges in comprehending mathematical notation (Hendrycks et al., 2021; Gong et al., 2022; Petersen et al., 2023; Dao & Le, 2023; Shen et al., 2023; Reusch et al., 2024; Qiao et al., 2024). These challenges stem from the complex formula structure, diverse formula representations, and ambiguous implicit semantics (Peng et al., 2021). For example, $x = \frac{-b \pm \sqrt{b^2 - 4ac}}{2a}$ involves nested operations, while different notations, such as $\frac{x}{y}$, $x/y$, $x \div y$, and $x \cdot y^{-1}$ can represent the same mathematical relationship, alongside the contextual meanings of symbols (e.g., $i$ as an index or imaginary unit) further complicate the understanding. These difficulties highlight the need for rich, specialized datasets to train models for mathematical content. However, existing datasets face scalability constraints due to expert curation or lack diversity in problem types and notation.

To address the need, we propose a framework, Math Mutator (MAMUT), for generating high-quality and diverse mathematical formulas to enhance the training and comprehension capabilities of mathematical language models. MAMUT allows for the creation of mathematically equivalent formulas (EquVG) and challenging non-equivalent ones that appear similar (FalseVG). This includes random alterations in variable and function identifiers and variations in LaTeX notation that leverage mathematical properties such as commutativity and symmetry. Additionally, we extend this approach to text containing mathematical LaTeX notation, ensuring consistent changes in identifiers and notation styles across textual contexts. We apply MAMUT to generate four datasets (see Figure 1 and Table 1) designed for the training of mathematical language models, e.g., for further mathematical pretraining on equation completion tasks. We apply this mathematical pretraining to several models and evaluate them on mathematical retrieval tasks, showing that models trained on MAMUT-enhanced data outperform existing mathematical models.

## 2 Related Work

This section covers language models, datasets, and data augmentation techniques in mathematical contexts.

### 2.1 Mathematical Language Models

The success of transformer-based models (Vaswani et al., 2017), such as Bidirectional Encoder Representations from Transformers (BERT) (Devlin et al., 2019), led to the development of domain-specific models, including SciBERT for scientific texts (Beltagy et al., 2019) and CodeBERT for programming (Feng et al., 2020). These models improve over general-purpose models by training on domain-specific data. Similarly, specialized models have been developed for mathematics, such as MathBERT (Shen et al., 2021),

| Query | Relevant Documents | Not Relevant Documents |
|:---:|:---:|:---:|
| $(a+b)^2 = a^2 + 2ab + b^2$ | $a^2 + 2ab + b^2 = (a+b)^2$ | $(a+b)^2 + a^2 = 2ab + b^2$ |
| | $(c+d)^2 = c^2 + 2cd + d^2$ | $(a+b)^2 = a^2 + 2ab + a^2$ |
| | $(a+b)^2 = a^2 + b^2 + 2ab$ | $(a+b)^2 = a^2 + b^2$ |
| $a^2 + b^2 = c^2$ | $c^2 = a^2 + b^2$ | $a^2 + b^2 + c^2$ |
| Pythagorean Theorem | $a^2 + b^2 = c^2$ | Pythagorean Identity |
| $\sum_{n=1}^{\infty} \frac{1}{n}$ | $\sum_{k=1}^{\infty} k^{-1}$ | $\sum_{n=1}^{\infty} \frac{1}{n^2}$ |
| $f'(x) = \lim_{h \to 0} \frac{f(x+h)-f(x)}{h}$ | $\frac{d}{dz}g(z) = \lim_{d \to 0} \frac{g(z+d)-g(z)}{d}$ | $\lim_{x \to 0} f(x) = 0$ |

Table 2: Examples for MIR queries including relevant and not relevant documents. Note that *Pythagorean Identity* $(\sin^2(x) + \cos^2(x) = 1)$ is not relevant for the query *Pythagorean Theorem* $(a^2 + b^2 = c^2)$.

MathBERTa (Novotný & Štefánik, 2022), and others (Peng et al., 2021; Reusch et al., 2022; Liu et al., 2023; Shao et al., 2024). They typically employ additional mathematical tokens and were pretrained on mathematical datasets (see Section 2.2).

A key application of mathematical language models is Mathematical Information Retrieval (MIR) (Dadure et al., 2024; Zanibbi et al., 2025), where the goal is to retrieve relevant documents based on a user's query, where both may contain mathematical content (see Table 2 for examples). Traditional MIR systems rely on keyword matching or simple embeddings (Kim et al., 2012; Greiner-Petter et al., 2020), while more sophisticated techniques leverage explicit mathematical knowledge, such as operator trees or formula unification (Kristianto et al., 2016; Mansouri et al., 2019; 2022b; Aizawa & Kohlhase, 2021; Peng et al., 2021). Transformer-based models offer new possibilities for MIR by addressing key challenges such as integrating natural and mathematical language, directly processing LaTeX input, and handling diverse notations. As a result, mathematical language models have been successfully adapted to MIR (Novotný & Štefánik, 2022; Reusch et al., 2022; Zhong et al., 2023).

Despite their promising performance, specialized mathematical models still face challenges in understanding mathematical notation (Gong et al., 2022; Shen et al., 2023), especially when it comes to handling variable names and recognizing mathematical equivalence beyond superficial textual similarities (Reusch et al., 2024). This motivates the creation of specialized datasets that reflect the unique roles of variables and aim to improve mathematical modeling.

## 2.2 Mathematical Datasets

The need for mathematical models has led to the development of various collections aimed at enhancing and evaluating language model capabilities in context of mathematics. Manually curated datasets like MATH (Hendrycks et al., 2021), GSM8K (Cobbe et al., 2021), and MathOdyssey (Fang et al., 2024) test problem-solving skills but are typically small, and reliant on expert input. Synthetically generated datasets like the Mathematics Dataset (Saxton et al., 2019), AMPS (Hendrycks et al., 2021), and HARDMATH (Fan et al., 2024) offer scalability but may lack diversity in problem topics, as they rely on generation rules, although they can produce a wide variety of similar but distinct problems (with changing numbers, symbols, ...), which can be beneficial for models learning to generalize across formula representations. Datasets like NTCIR (Zanibbi et al., 2016) and ARQMath (Mansouri et al., 2022a), sourced from the existing repositories arXiv, Wikipedia, and the Mathematical Stack Exchange, provide a broad range of real-world mathematical problems. However, they lack controlled variations of specific formulas, an important aspect for training MIR models to classify whether two symbolic representations are mathematically equivalent.

|  | Natural Language | Mathematical Language |
|---|---|---|
| **Purpose** | General human communication, including opinions and feelings | Precise description of mathematical concepts |
| **Vocabulary** | Large set of words (language dependent), sometimes with ambiguous meaning (e.g., *love*, *happy*, *data*) | Small set of well-defined symbols (e.g., $x$, $+$, $\mathbb{R}$, sin, $\forall$, $\infty$, $\int$) and terms (e.g., *Eigenvalue*, *Derivative*, *Field*) with precise meanings |
| **Grammar** | Rather flexible | Strict rules |
| **Clarity** | Often imprecise, open to interpretation | Single, unambiguous interpretation |
| **Evolution** | Evolves over time naturally, new words, phrases, and idioms emerge or disappear | Evolves slower, changes are introduced by mathematicians and are backward compatible |
| **Writing Style** | Linear structure in sentences and paragraphs using standard formats | Requires specialized formats (e.g., LaTeX) to represent complex notation in a structured way |

Table 3: Comparison of natural and mathematical language.

### 2.3 Mathematical Data Augmentation Techniques

Recent advancements in mathematical data augmentation have introduced various innovative methods aimed at enhancing the diversity and depth of training materials. InfinityMath (Zhang et al., 2024) utilize GPT-4 (Achiam et al., 2023) to transform specific mathematical problems into generic templates. These templates can then generate multiple variations of the original problem, altering numerical values or structural complexity, thereby increasing the dataset's variety. Similarly, Li et al. (2024) propose a method to formalize mathematical problems written in natural language, alter the difficulty by adjusting the problem's operations, and then informalize these changes back into natural language using GPT-4, preserving mathematical integrity across different levels of complexity. MathGenie (Lu et al., 2024) augments step-by-step solutions by generating modified candidate solutions with a Llama model (Touvron et al., 2023) with verified correctness, and then back-propagating these solutions to a modified question. You et al. (2024) augment data by applying different strategies, including rephrasing and reorganization with LLM, and question alteration.

These approaches primarily focus on diversifying problem content rather than varying mathematical notation (e.g., $(a + b)^2 = a^2 + 2ab + b^2$ vs. $(x + y)^2 = x^2 + y^2 + 2yx$). In contrast, Reusch et al. (2024) explore variable renaming in training data to prevent models from taking shortcuts in problem-solving, such as relying only on variable overlap. Building on this idea, our study enhances mathematical formula diversity through substitutions and other techniques.

## 3 Natural and Mathematical Language

Mathematical language differs fundamentally from natural languages such as English, German, or Chinese. While natural language is used for general communication and often conveys subjective information, mathematical language serves a highly specialized purpose to precisely describe mathematical topics, such as definitions, theorems, and proofs. It consists of both symbolic expressions (e.g., $a^2 + b^2 = c^2$ and $\int_a^b \sin(x), dx$) and specialized terminology (e.g., *derivative* and *eigenvalue*). Despite their differences, natural and mathematical languages share some structural similarities. Both use symbols arranged in a syntax that conveys meaning, and both can be represented in textual form. However, there are some key differences (Ilany et al., 2010; Scarlatos & Lan, 2023) summarized in Table 3. A crucial challenge for mathematical language models is *symbol abstraction*. In mathematical expressions, certain symbols act essentially as wildcards and can be replaced without changing the mathematical meaning. These symbols are either *variables* (e.g., $x$, $\alpha$, $A$) or *generic functions* (e.g., $f$, $g$ or $\varphi$), i.e., functions not tied to a specific mathematical object (e.g., Euler's Gamma function $\Gamma(z)$). For example, a model should recognize that the first binomial formula,

$$(a + b)^2 = a^2 + 2ab + b^2, \tag{1}$$

| Names | Factorial, Definition of a factorial |
|---|---|
| **Version 1** | $n! = 1 \cdot 2 \cdot \ldots \cdot n$ |
| **Version 2** | $n! = \prod_{i=1}^{n} i$ |
| **Version 3** | $\forall n \in \mathbb{N} : (n+1)! = (n+1) \cdot n!, \ 0! = 1$ |
| **Version 4** | For any natural number $n$, we have $n!$ is defined as $n! \coloneqq \prod_{i=1}^{n} i$. |
| **Version 5** | For any natural number $n$, $n!$ can be obtained by multiplying all natural numbers from 1 to $n$ together. |
| **Similar Formula** | Binomial Coefficient Formula |
| **False Version 1** | $n! = 1 \cdot 2 \cdot 3 \cdot 4 \cdot n$ |
| **False Version 2** | $\forall n \in \mathbb{N} : (n+1)! = (n-1) \cdot n!, \ 0! = 0$ |
| **Falsifying Replacements** | $\prod \rightarrow \sum, \mathbb{N} \rightarrow \mathbb{R},$ "natural" $\rightarrow$ "real" |

Table 4: Example entry for the definition of a factorial from the NMFT dataset (partially).

is equivalent to $(c+d)^2 = c^2 + 2cd + d^2$, despite different variable names. In contrast, $(a+b)^2 = a^2 + 2ab + \boldsymbol{a^2}$ uses the same variables but in a mathematically non-derivable way. Likewise, the modified formula $(a+b)^2 = a^2 + b^2 + 2ab$ appears different from Eq. (1), but it is, in fact, mathematically equivalent.

Another important aspect is the structure of mathematical formulas. Consider the two formulas $2^x$ and $x^2$. Assuming a character-wise LaTeX tokenization, both formulas use the same tokens but in a different order. A model should not treat $x^2$ as equivalent to $2^x$ (but instead to $x \cdot x$). These structural variations highlight the need for a model that goes beyond simple token matching and actually understands mathematical meaning. Transformer language models (Vaswani et al., 2017) have shown their capabilities in modeling natural language, hence, it is worth exploring their potential to precisely capture mathematical language.

## 4 Named Mathematical Formula Templates (NMFT)

Previous datasets provide formulas and mathematical texts with significant variance across a wide range of mathematical topics. For the purpose of our proposed data augmentation methods introduced in the next section, it is necessary to parse formulas into symbolic expressions. Real-world datasets contain formulas with various notations, some of them might be parsed incorrectly, or the parsing even fails completely. Therefore, we created a dataset consisting of only a few but high-quality parsable formulas. This dataset includes 71 well-known mathematical identities that are easily recognizable and associated with one or multiple names. For example, $a^2 + b^2 = c^2$ represents the Pythagorean theorem, while $(a+b)^2 = a^2 + 2ab + b^2$ represents the first binomial formula (Eq. (1)). Since the mathematical formulas are associated with their names, we call this dataset Named Mathematical Formula Templates (NMFT), as the formulas serve as templates for deriving modified versions. An example entry can be found in Table 4, and Table 7 lists all identities.

Each identity provides multiple representations, such as $\forall a, b \in \mathbb{R} : (a+b)^2 = a^2 + 2ab + b^2$ as another, more detailed version of the first binomial formula. Additionally, some representations are provided as descriptive text, e.g., *"In a right-angled triangle with side lengths a, b, and c, where c represents the length of the hypotenuse, the equation $a^2 + b^2 = c^2$ holds true"*. Others paraphrase formulas textually, e.g., *"$a^2 + b^2$ is equal to $c^2$"*, reinforcing associations between the equals sign $=$ and *"equals"*. These textual versions intentionally exclude the name of the identity to make MIR tasks harder, where the name serves as the query. For each provided identity version, the variables and function symbols are explicitly given to assist the parsing and version generation. To enhance the generation of challenging falsified versions, similar-looking formulas are provided (e.g., the first binomial formula for the Pythagorean theorem, as both identities contain

multiple powers of two), or hints to falsify any given representation by a string replacement (e.g., removing *"right-angled"* to falsify the previous descriptive example of the Pythagorean theorem). The descriptive text versions have been partially generated by using GPT-3.5 (Brown et al., 2020) and all manually verified for validity. Typically, we call entries of NMFT and of datasets generated from it *formulas*, but both, pure mathematical formulas and textual descriptions of formulas are meant.

## 5 Math Mutator (MAMUT)

Our goal is to create high-quality, large, and diverse mathematical datasets to enhance mathematical modeling. We introduce Math Mutator (MAMUT), a framework consisting of two core algorithms designed to generate both equivalent and falsified versions of a given formula. The first algorithm, Equivalent Version Generation (EquVG), presented in Section 5.1, automatically generates various versions of a given formula, expanding the training data and enabling the model to learn math-specific language rules, such as treating variables as placeholders that can be substituted without changing the validity of an expression. The second algorithm, Falsified Version Generation (FalseVG), introduced in Section 5.2, slightly modifies formulas to create mathematically non-equivalent versions of the original formula, offering challenging negative examples for MIR tasks.

### 5.1 EquVG: Variations of Mathematical Formulas

The key idea of this section can be summarized as follows: Given a mathematical formula, our aim is to generate mathematically equivalent variations of this formula, called *equivalent versions*. For instance, consider the formula

$$(a + b)^2 = a^2 + 2 \cdot a \cdot b + b^2. \tag{2}$$

In this context, we observe that all the following formulas describe the same mathematical relationship, namely the first binomial formula:

$$(b + a)^2 = a^2 + b^2 + 2 \cdot b \cdot a, \tag{3}$$

$$(a + b)^2 = a \cdot a + 2 \cdot a \cdot b + b^2, \tag{4}$$

$$a^2 + 2 \cdot a \cdot b + b^2 = (a + b)^2, \tag{5}$$

$$(c + d)^2 = c^2 + 2 \cdot c \cdot d + d^2, \tag{6}$$

$$(\lambda + Z)^2 = \lambda^2 + 2 \cdot \lambda \cdot Z + Z^2. \tag{7}$$

By applying both additive and multiplicative commutativity, one can derive Eq. (3) from Eq. (2). In Eq. (4), the exponentiation $a^2$ is replaced by its definition $a \cdot a$. Since equality is a symmetric relation, equations remain valid after interchanging the sides, as done in Eq. (5). The final two equations can be obtained from Eq. (2) by substituting variables. This section is dedicated to the automated generation of such equivalent versions. Note that for a complete mathematical expression, it would be necessary to specify the range of values (e.g., of variables) for which the statement holds. For example, a complete expression of Eq. (2) could be $\forall a, b \in \mathbb{R} : (a + b)^2 = a^2 + 2 \cdot a \cdot b + b^2$. However, in practical applications like MIR systems, the shortened version Eq. (2) may also be used, for the sake of simplicity. Therefore, the somewhat less precise mathematical formulations in Eqs. (2)-(7) are often sufficient.

The complete workflow of our proposed Equivalent Version Generation (EquVG) algorithm is depicted in Figure 2. The input consists of a formula written in LaTeX format. To implement transformations, as seen in Eqs. (2)-(7), we can identify two steps: the substitution of symbols and the modification of the mathematical notation. For both of these purposes, it is helpful to represent the formula not as a string but as a structured data format that captures the mathematical relationships and dependencies. This representation is achieved by parsing the LaTeX formulas into a symbolic expression format, essentially creating an operator tree. The symbolic representation categorizes elements into numbers, variables, functions, and other mathematical objects, establishing a structured relationship between them. This organization enables the identification and substitution of variables (x, \alpha, ...) and generic functions (f, g, ...) to derive a mathematically

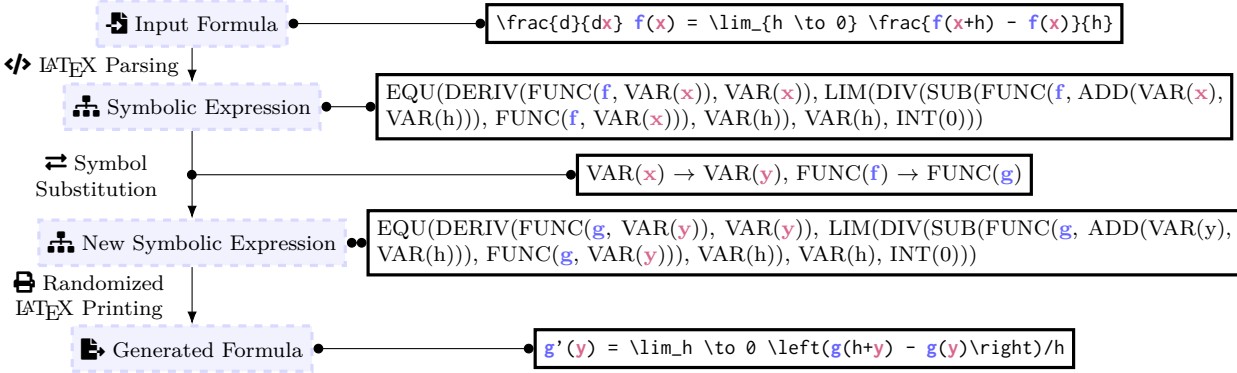

Figure 2: Visualization of the EquVG algorithm.

equivalent representation using different symbols. This substituted expression is then converted back into LaTeX format during the printing process, which includes the desired modifications of mathematical notation, such as writing $a \cdot a$ instead of $a^2$. In the following, we will provide a more detailed explanation of the three steps of EquVG: parsing, substituting, and printing.

**</> LaTeX Parsing**   Parsing a LaTeX formula into a symbolic expression presents certain challenges. For instance, if a letter precedes parentheses, it can be interpreted either as a multiplication (with omitted multiplication symbol, e.g., $v(x + y) = v \cdot (x + y)$) or as a function call ($v(x + y)$). The symbol $e$ could be either a variable or Euler's number $e \approx 2.718$. Likewise, the symbol $i$ might function as a variable (e.g., as a summation index in $\sum_{i=1}^{n} i^2$) or as the imaginary unit, sometimes expressed in LaTeX as i (\mathrm{i}) to avoid ambiguity. It is crucial for our purposes to determine whether a symbol is a substitutable variable or a constant. Otherwise, the imaginary unit might be incorrectly substituted by another symbol, resulting in a non-equivalent expression. We have addressed this issue, partially by applying heuristics that consider the context ($i = 1$ within the formula indicates a variable, while $i\pi$ indicates the imaginary unit, as in $e^{i\pi}$). Furthermore, we introduce a safeguard to handle cases where the parser is uncertain about whether to treat a symbol as a variable or the imaginary unit. In these cases, the symbol is represented in a way that prevents substitution while maintaining its appearance as the plain $i$, without enabling complex unit formatting options. Despite these measures, parsing can still fail in cases with unusual or malformed notation.

The formula parsing can be conceptually expanded to include the parsing of text containing LaTeX formulas. Such texts are referred to as *mathematical texts* in this study. The text parts remain unchanged during substitution and printing, only the formula parts are processed consistently by EquVG as shown in Figure 3. This allows to consistently change symbols in a mathematical text throughout all its formulas. Within a mathematical text, formulas are defined as text in between dollar signs ($...$), as used in LaTeX documents to write inline mathematical formulas.

**⇄ Symbol Substitution**   The symbolic expression format allows the substitution of symbols by simply replacing all occurrences of a given symbol within the expression. This is conceptually related to $\alpha$-conversion in definitional equality (Nederpelt & Geuvers, 2014), where renaming of bound variables (e.g., $k$ in $n := \prod_{k=1}^{n} k$) preserves the meaning of an expression. Our framework generalizes this to also include free variables (e.g., $n$ in the same factorial definition).

The generation of a substitution (i.e., a mapping of symbols) involves two steps. Firstly, a subset of all symbols in the expression is randomly selected. Secondly, a new symbol is chosen for each selected symbol. The aim of the substitution process is to generate diverse formulas that are similar to formulas occurring in real-world scenarios. It is important to note that, intuitively, Eq. (6) with variables $c$ and $d$ appears more familiar for a binomial formula than Eq. (7), which uses Greek and uppercase Latin letters ($\lambda$ and $Z$) that are not commonly used as variables in the context of binomial formulas. When variables are selected entirely at random from a uniform distribution over all Latin and Greek letters, unfamiliar symbol usage is more likely. This motivates the introduction of *symbol groups*, which categorizes similar variables or functions together. The defined symbol groups can be found in Table 11, along with a description of a typical mathematical

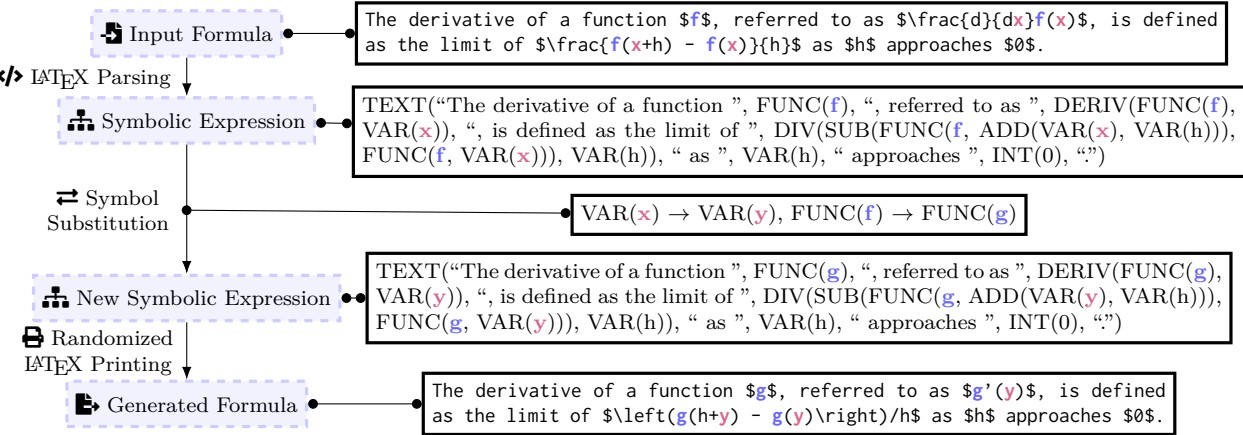

Figure 3: Visualization of the EquVG algorithm for a mathematical text.

context for each group. For instance, we have the group of indices $\{i, j, k, l\}$ and group of vectors $\{u, v, w\}$. It is worth noting that variables can belong to multiple groups. Given a symbol that should be substituted with a new symbol, all symbols from the relevant symbol group(s) are candidates. Additionally, the most common variable $x$ is a candidate in every variable group to reflect its common usage. To add variety, a random symbol may be added by chance to the set of candidates, selected from commonly used lowercase or uppercase Latin letters or lowercase Greek letters. Symbols that refer to constants in certain contexts, such as $e$, $i$, and $\pi$, are excluded. Given the set of candidates, a random candidate is chosen. However, it is sometimes necessary (or at least useful) to make the symbol selection dependent on multiple substitution symbols. For instance, consider the Fundamental Theorem of Calculus: $\int_a^b f(x)\, dx = F(b) - F(a)$. Here, we have two generic functions, $f$ and $F$. Whenever a symbol has related variants in the formula, such as uppercase and lowercase forms or corresponding Greek letters (e.g., $a$, $A$, $\alpha$), the algorithm automatically preserves these relationships by restricting possible substitutions accordingly. For instance, substituting $f$ and $F$ by $g$ and $G$, respectively, yields an equivalent version $\int_a^b g(x)\, dx = G(b) - G(a)$. Again, this is rather mathematically imprecise but aligns with the implicit assumptions on mathematical notation found in real-world datasets. Another possible generated version is $\int_{a_1}^{a_2} f(x)\, dx = F(a_2) - F(a_1)$ where $a$ and $b$ are substituted by indexed variables $a_1$ and $a_2$ respectively. In cases where multiple variables of the same variable group appear in the same formula, the generation algorithm may randomly perform such indexed substitutions. The indexing enforces the model not only to attend to the variable itself but also its modifications, in this case, its index.

🖨 **Randomized LATEX Printing** In the final step of EquVG, the symbolic expression is converted back into LATEX format. To further increase the variety of generated formulas, the LATEX printer makes randomized printing decisions. These variations can be categorized into two main sources: mathematical and LATEX notation. The parsed and printed formulas in Figure 3 are illustrating these differences. For example, the input text used the explicit notation for a derivative, \frac{d}{dx}f(x), while the printed substituted expression uses the shorthand notation g'(y). We developed a list of equivalent mathematical notations, where the printer randomly selects one of the available ones for printing. As another example in Figure 3, instead of the fraction notation with \frac, the printer used the forward slash / to denote division. Since addition is commutative, h+y is printed instead of y+h. These examples represent mathematical variations, as they express a mathematical concept in an equivalent way. In contrast, the usage of \left and \right commands represents LATEX variations, since these commands are not essential for mathematical reasons but only for a differently rendered text. In addition to the already covered examples, the randomization of the LATEX printer includes the notation of

- equalities (x = y vs. y = x),

- inequalities (x > 0 vs. 0 < x),

- multiplication symbols (a \cdot b vs. a * b vs. a \times b vs. ab),

- divisions (`2/n` vs. `2 \cdot n^{-1}` vs. `\frac{2}{n}` vs. `\frac2n`),

- integer powers (`a^3` vs. `a^2\cdot a` vs. `a \cdot a \cdot a`),

- inverse trigonometric functions (`\asin(x)` vs. `\arcsin(x)` vs. `\sin^{-1}(x)` vs. `(\sin(x))^{-1}`),

- higher order derivatives (`f'''(x)` vs. `f^{(3)}(x)` vs. `\frac{d^3}{dx^3} f(x)`),

- expected values (`\mathbb{E}[X]` vs. `\operatorname{E}[X]` vs. `E[X]`),

- matrix determinants (`\det(A)` vs. `|A|`),

- binomial coefficients (`\binom{n}{k}` vs. `{n \choose k}`),

- empty sets (`\emptyset` vs. `\varnothing` vs. `\{\}`), and

- natural logarithms (`\ln(x)` vs. `\log_e(x)`).

As real-world data uses different styles of notations, language models should be capable of understanding all commonly used notations. This is similar to the notion of synonyms in natural language. The randomized LaTeX printing provides an automation to diversify training data, such that models can learn the different notations. The combination of parsing, substituting, and printing results as part of EquVG is a powerful tool to increase the training data size significantly. Additionally, research has shown that using training data with substituted query-document pairs for MIR helps the model to less focus on shallow features such as variable overlapping (Reusch et al., 2024), confirming the usage of substitutions in EquVG.

## 5.2 FalseVG: Generating Challenging Negative Examples

We believe that a classification task determining whether two formulas describe the same mathematical concept helps the model to encode mathematics more effectively. To train models on such a task, we require both positive and negative formula pairs, similar to a MIR training. While positive pairs are often readily available in datasets, identifying meaningful negative pairs is more challenging, as datasets rarely contain explicit negative examples.

A common approach to extract negative pairs is random sampling from datasets by pairing two random formulas. However, this may lead to simplified feature extractions. For instance, the model might learn to simply check for the presence of an important function, like whether both formulas contain the determinant function `\det`. Given a random negative document, a naive classifier that checks if a determinant is part of the formula would likely perform well, due to the rarity of the determinant function across most mathematical contexts. The language model may adapt to this behavior during training.

To prevent models from learning such easy shortcuts instead of the true semantic understanding, researchers have successfully used challenging negative examples in other domains (Cai & Liu, 2020; Qiu et al., 2021). We introduce the Falsified Version Generation (FalseVG) algorithm to generate *falsified versions* of a given formula, meaning a similar-looking but *not* mathematically equivalent formula. Since the formulas are already parsed into a symbolic expression for EquVG, we can simply use and modify this representation. We have developed and implemented eight modification strategies, which are described below. Table 10 provides illustrative examples of each strategy. Similar to EquVG, these strategies can also be applied to mathematical texts, by applying the strategies to the text's formulas.

**⚌ Equality**   Falsifying an *equality* can be achieved by inserting or removing a term on one side of the equality. This can be done either at the outermost level (e.g., changing $\sin(x) = \ldots$ to $\sin(x) + 1 = \ldots$) or within a sub-expression (e.g., changing $\sin(x) = \ldots$ to $\sin(x + 1) = \ldots$). When inserting a term, the algorithm selects either a subexpression from the entire formula, a random new variable, or a random number. Importantly, the algorithm avoids modifications that will not change the validity of an equality, such as adding zero or multiplying by one. This strategy enforces the model to focus on the entire formula and long-term dependencies.

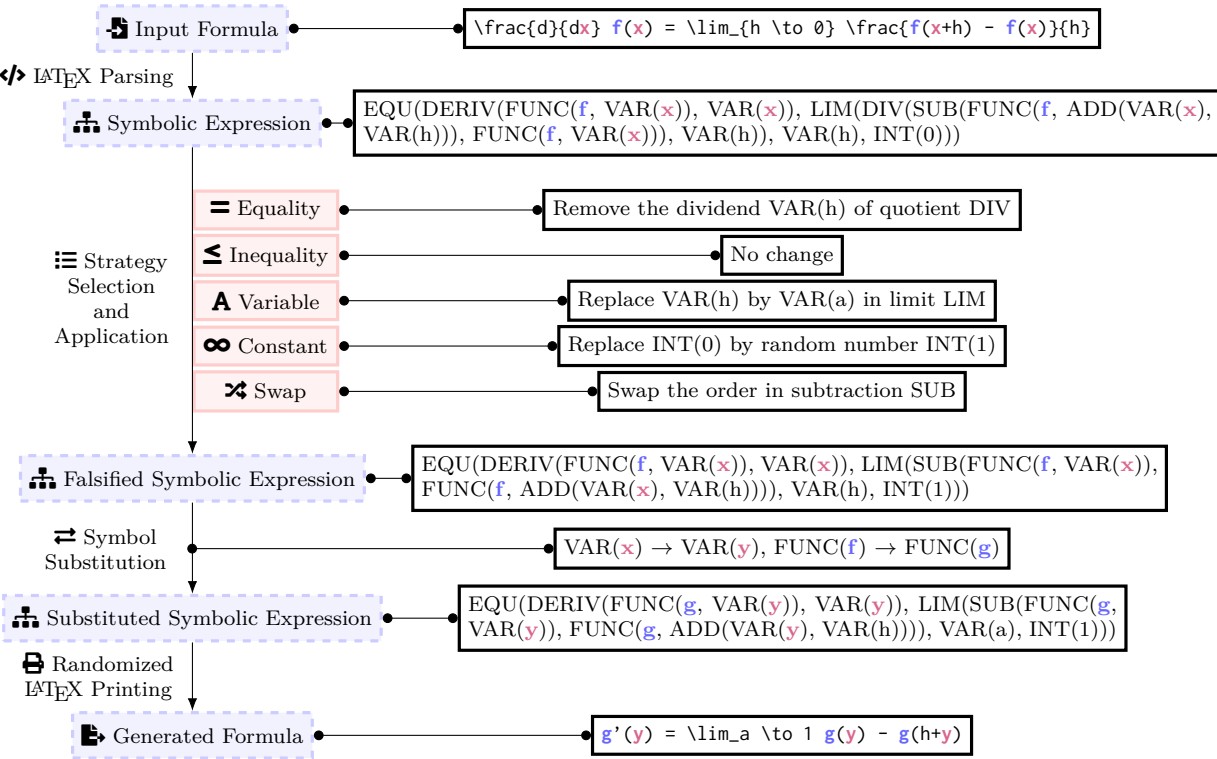

Figure 4: Visualization of the FalseVG algorithm.

**≤ Inequality** To falsify an *inequality*, we simply invert the inequality symbol. Thus, the symbol $\leq$ is replaced by $>$ and vice versa. The same replacement holds for $\geq$ and $<$. The not equals symbol $\neq$ can be replaced by $=$, but not vice versa (as $=$ indicates an equality). Similarly to the strategy equality, the model is forced to encode long-term dependencies using this strategy.

**⤬ Swap** The strategy *swap* involves altering unary and binary functions. Unary functions such as sine, square root, or logarithm get replaced by different random unary functions. In the case of binary non-commutative functions, we swap the order of the two arguments. These non-commutative functions are subtraction, division, and exponentiation (e.g., $x^2$ becomes $2^x$). These changes enforce the model to rely on the order of operands rather than just token occurrences in a random order.

**A Variable** The strategy *variable* essentially aims to split a single variable (e.g., $a$ in $(\boldsymbol{a}+b)^2 = \boldsymbol{a}^2+2\boldsymbol{a}b = b^2$) into two (e.g., into $a$ and $c$ in $(\boldsymbol{c}+b)^2 = \boldsymbol{a}^2 + 2\boldsymbol{c}b + b^2$). Specifically, if a variable occurs at least twice in the formula, it might be randomly replaced by another variable for a proper and nonempty subset of its occurrences (i.e., at least one occurrence is replaced and at least one occurrence remains unchanged). This strategy enforces the model to check for a consistent use of symbols in the entire formula.

**∞ Constant** The strategy *constant* focuses on numbers ($1$, $2$, $e$, $\pi$, $\infty$, ...) as well as variables that are typically considered to be constant within an expression, such as the upper limit $n$ of an indexed sum like $\sum_{i=1}^{n} i^2$. These constants are replaced by other constants enforcing the model to learn what tokens a certain formula should contain.

**ⵖ Distribute** The strategy *distribute* is inspired by the distributive law, a fundamental mathematical rule relating two binary functions. A standard example for real numbers is that multiplication distributes over addition since $x \cdot (y+z) = x \cdot y + x \cdot z$ holds for all real numbers $x, y, z$. This rule motivates this strategy, which applies a modified distributive law to non-distributive functions. Specifically, for a unary function $f$ and a binary function $\oplus$ in infix notation, the relation $f(x \oplus y) = f(x) \oplus f(y)$ is (falsely) assumed. We use addition and multiplication as binary functions and the logarithm, factorial, power with fixed base, and trigonometric functions for the unary function. This readily results in examples where commonly known identities are

falsified, e.g., the falsified product of powers rule is $2^x \cdot 2^y = 2^{x \cdot y}$ (instead of the correct $2^x \cdot 2^y = 2^{x+y}$). The falsified sine additivity yields $\sin(x+y) = \sin(x) + \sin(y)$ (instead of $\sin(x+y) = \sin(x)\cos(y) + \cos(x)\sin(y)$). This strategy enforces the model to notice the presence of parantheses and to enhance its understanding of operator relationships, including precedence.

**👥⚙ Manual**    While all previous strategies focused on modifying a formula by applying generally valid transformation rules to falsify it, this strategy relies on *manual* transformation or replacement rules. These rules refer to the specifically newly created NMFT dataset (see Section 4). The rules can be explicitly given falsified versions (e.g., $\forall n \in \mathbb{N} : n! = 1 \cdot 2 \cdot n$), references to different but similar formulas (e.g., law of cosines for the Pythagorean theorem), or falsifying replacement rules. For example, a formula replacement might change $\forall n \in \mathbb{N}$ to $\forall n \in \mathbb{R}$ if the quantified term only holds for natural but not for real numbers. These rules are also applicable to mathematical texts, where, for instance, *natural* can be replaced by *real*.

**🎴 Random**    The simplest approach to generate a falsified formula is to use a *random* formula, meaning an earlier generated equivalent version of a different formula. This approach is especially important to increase the models' robustness in real-world applications, where most of the input pairs are not inherently challenging.

The complete FalseVG algorithm is summarized in Figure 4. It involves applying a random subset of the strategies to a parsed symbolic expression. Note that some strategies are not applicable to certain formulas, resulting in no changes. However, if at least one strategy succeeds, a falsified symbolic expression is generated. Finally, a random symbol substitution and randomized LaTeX printing are performed to create the final formula as a string, identically to EquVG.

# 6    Generated Datasets

This section presents four datasets generated using MAMUT employing EquVG and FalseVG. Our implementation, built on SymPy (Meurer et al., 2017), is detailed in Appendix B.1. Table 5 summarizes key statistics of the generated datasets, including Hugging Face identifiers, while Appendix B.2 reports example entries. All entries ensure uniqueness at the string level. Two of the generated datasets (NMF and MFR) are based on the specifically created NMFT dataset (see Section 4), while the other two datasets (MF and MT) are derived from two existing diverse sources that combine natural language with mathematical notation: Answer Retrieval for Questions on Math (ARQMath) (Mansouri et al., 2022a) and the Khan Academy problems in Auxiliary Mathematics Problems and Solutions (AMPS) (Hendrycks et al., 2021). While we focus on these sources, MAMUT is applicable to any mathematical corpus containing LaTeX notation. ARQMath, sourced from the Mathematics Stack Exchange, benefits from a user-rating system that ensures high-quality discussions and problem-solving content. The Khan Academy problems in AMPS provide structured exercises used for educational purposes. Example dataset entries are shown in Appendix A.

**Mathematical Formulas (MF)**    This dataset consists exclusively of mathematical formulas extracted from AMPS and ARQMath, enriched with variations by the EquVG algorithm. However, not all formulas from these raw datasets are included in the MF dataset. Only formulas are selected being suitable for a Masked Language Modeling (MLM) task (Devlin et al., 2019), where a masked token's value can be concluded by the remaining context of the formula. For example, a masked formula such as $\pi > $ `[MASK]` has infinite algebraic solutions, like 3, 0, or any other value that can fill the masked position in a mathematical valid sense. Therefore, the formulas are restricted to equalities and implications to ensure meaningful inferences. Additionally, only generally valid formulas according to SymPy's formula solver (e.g., $\tan(x) = \frac{\sin(x)}{\cos(x)}$ as this equation holds for all $x \in \mathbb{R}$) are considered as input formulas for EquVG to ensure high data quality. In case of an equation without general validity (e.g., $x^2 = 2$) but with existing solution(s) found by SymPy, the equation can be transformed into an implication (e.g., $x^2 = 2 \Rightarrow x = -\sqrt{2}$ or $x = \sqrt{2}$). A few examples of extracted formulas are (original LaTeX formatting is preserved):

$$\tan(x) = \sin(x)/\cos(x),$$
$$-\frac{2}{5} \div -\frac{1}{6} = -\frac{2}{5} \times -\frac{6}{1},$$
$$3x = 210 \Rightarrow x = 70,$$

| Name | Hugging Face Identifier | Original Dataset(s) | Raw Entries | Generated Versions | ∅ v.p.f. | Max v.p.f. |
|------|------------------------|---------------------|-------------|--------------------|----------|------------|
| MF | ddrg/math_formulas | AMPS | 30,985 | 958,735 | 30.9 | 101 |
| | | ARQMath | 55,894 | 2,257,826 | 43.3 | 101 |
| | | Both | 82,765 | 3,198,108 | 38.6 | 101 |
| MT | ddrg/math_text | AMPS | 62,099 | 2,542,015 | 40.9 | 101 |
| | | ARQMath | 690,333 | 4,480,369 | 6.5 | 96 |
| | | Both | 752,428 | 7,022,384 | 9.3 | 101 |
| NMF | ddrg/named_math_formulas | NMFT | 71/ 522 | 23,707,392 | 333,906 | 400,000 |
| MFR | ddrg/math_formula_retrieval | NMF | 71/ 522 | 23,702,560 | 334,092 | 400,000 |

Table 5: Summary of the generated datasets. The abbreviation *v.p.f.* stands for *versions per formula*. For MF, the *Generated Versions* values do not sum up from AMPS and ARQMath to *Both* due to duplicate removal. The raw values of NMF and MFR refer to the number of mathematical identities and the total number of provided version templates of these identities, respectively.

$$\frac{3}{13} - \frac{2}{13} = \frac{1}{13},$$
$$e^{2\pi i} = (e^{\pi i})^2 = (-1)^2 = 1,$$
$$\sqrt{25} = 5,$$
$$(n+1) \times (n-1)! = \frac{(n+1) \times n \times (n-1)!}{n} = \frac{(n+1)!}{n}.$$

**Mathematical Texts (MT)**    While the previous dataset MF focuses exclusively on mathematical formulas, MT focuses on the relationship between mathematical formulas and natural language. Similarly to MF, MT is generated using the AMPS and ARQMath datasets, along with applying EquVG, which consistently changes variable names across the text and prints the LaTeX formulas in different ways. We only consider texts containing at least five formulas. Questions and answers of ARQMath are treated as separate text, while the AMPS data is treated as a single text where question and hints are concatenated. To ensure high data quality, only answers from ARQMath with at least five upvotes are included. We generate up to 100 additional versions for each suitable input.

**Named Mathematical Formulas (NMF)**    This dataset associates the name of a mathematical identity with either its formula or a describing text. It is derived from NMFT by applying both, EquVG and FalseVG, resulting in diverse positive and negative pairs. This data could be used to train a classifier that predicts whether a formula is a valid representation of an identity's name, using a Next Sentence Prediction (NSP)-like task (Devlin et al., 2019). In a typical NSP task, each positive pair is matched with a random negative pair, which changes when the positive pair is reused. To enhance training, we create an imbalanced dataset with four times more negative than positive pairs. This allows for training where positive pairs remain unchanged across epochs, while negative pairs vary between epochs (and remain challenging). With a maximum of four epochs, the model encounters unique negative pairs in each iteration. NMF originates from 71 mathematical identities, each with multiple base versions used to generate up to 400k versions per identity. About 60% of the NMF entries are textual descriptions, and the rest are pure mathematical formulas. For 20 of the 71 mathematical identities, fewer than 400k versions exist, as they offer fewer possibilities for generating versions, such as limited substitution options or fewer opportunities for creating randomized LaTeX.

**Mathematical Formula Retrieval (MFR)**    This dataset consists of formula pairs, classified as either mathematical equivalent or not. It is constructed by pairing each true formula version from NMF with an equivalent version and four falsified versions of that identity, all randomly sourced from NMF. This approach

preserves the positive-to-negative pair ratio while ensuring that negative pairs remain challenging. MFR can be used to train a MIR system for querying relevant formulas based on a similar formula, like a NSP task.

# 7 Experiments

In this section, we evaluate the effectiveness of MAMUT-generated data through mathematical pretraining and fine-tuning based on the four generated datasets introduced in the previous section. Our goal is to demonstrate that MAMUT enhances mathematical encodings, even when applied to models already pretrained on mathematical corpora.

## 7.1 Setup

We consider the BERT-base model (Devlin et al., 2019) as a general-purpose baseline and evaluate two math-specific variants initialized from BERT: MathBERT (Shen et al., 2021), trained on data from mathematical curricula, textbooks and arXiv, and Math-Pretrained-BERT (MPBERT) (Reusch et al., 2022), trained on AMPS and ARQMath (see Table 16 for model checkpoints). We focus on BERT-based models as transformer encoders remain a standard and effective choice for information retrieval tasks (Wang et al., 2024; Warner et al., 2024). We further pretrain these base models on MAMUT-enhanced datasets using two types of objectives: MLM on MF and MT, and NSP on NMF and MFR. Models pretrained on all four MAMUT-enhanced datasets are written with MAMUT as prefix (e.g., MAMUT-BERT). All models are subsequently fine-tuned on MIR tasks derived from NMF and MFR. These tasks are framed as query-based retrieval: given a name or a formula as query, the task is to retrieve matching formulas from a candidate set (e.g., retrieving the formula $(a + b)^2 = a^2 + 2ab + b^2$ from the queries *Binomial Formula* and $c^2 + 2 \cdot c \cdot d + d^2 = (c + d)^2$). We sample 250 positive training examples per mathematical identity of NMFT and ten times as many negatives, preserving the train-test split from pretraining. We evaluate using both binary classification metrics (precision (P), recall (R), F1) and standard IR ranking metrics: Precicion at $k$ (p@$k$), Average Precision (AP), and nDCG, that are averaged over all test queries, with higher values indicating better performance (Manning, 2009; Radlinski & Craswell, 2010). To ensure robust results, we further average across five fine-tuning runs. To allow an unbiased comparison where no task or data similar to the test data used during pretraining, we additionally pretrain models only on the MLM tasks MF and MT, excluding NMF and MFR used for fine-tuning. We denote these models with suffix MLM (e.g., MAMUT-BERT-MLM).

## 7.2 Results

Table 6 summarizes the main results, while implementation details and additional results are provided in Appendix C. Results of the fully pretrained models, shown in gray for completeness, are included despite their unfair advantage from being trained on more data during pretraining. Note that the fine-tuning test data was not seen during pre-training, and a new classification head was used during fine-tuning. Notably, the models pretrained only on MF and MT already outperform all baseline models across most metrics. Interestingly, even models that are already mathematically pretrained benefit from additional MAMUT-based training. This suggests that MAMUT introduces complementary patterns not present in prior mathematical corpora used for training. Remarkably, MPBERT mostly outperforms MathBERT, although the latter is often seen as the SoTA mathematical BERT-based model in recent studies (Scarlatos & Lan, 2023; Duan et al., 2024; Horowitz & Hathaway, 2024; Meadows et al., 2024). Overall, the results support the hypothesis that training on MAMUT-enhanced datasets improve mathematical capabilities of language models. Our best-performing models are those based on MPBERT, further pretrained with MAMUT-enhanced datasets. We publish the fully pretrained models on Hugging Face, see Table 17.

# 8 Discussion

## 8.1 Comparison to Other Mathematical Datasets

While existing mathematical datasets such as ARQMath (Mansouri et al., 2022a), AMPS (Hendrycks et al., 2021), MATH (Hendrycks et al., 2021), and GSM8K (Cobbe et al., 2021), focus primarily on problem diver-

| Model | NMF-FT | | | | | | | MFR-FT | | | | | | |
|---|---|---|---|---|---|---|---|---|---|---|---|---|---|---|
| | P | R | F1 | p@1 | p@10 | AP | nDCG | P | R | F1 | p@1 | p@10 | AP | nDCG |
| BERT | 49.2 | 98.0 | 65.5 | 65.6 | 62.0 | 69.7 | 84.3 | 39.8 | 94.9 | 56.1 | 58.9 | 58.8 | 57.5 | 82.9 |
| MathBERT | 58.7 | 99.0 | 73.7 | 87.1 | 73.7 | 83.8 | 92.3 | 43.5 | 95.8 | 59.8 | 78.6 | 75.1 | 69.4 | 89.1 |
| MPBERT | 66.8 | 99.5 | 79.9 | 83.7 | 74.2 | 84.1 | 91.9 | 59.0 | 99.0 | 74.0 | 79.4 | 77.4 | 76.4 | 91.2 |
| MAMUT-BERT-MLM | 74.4 | 98.9 | 85.0 | **89.7** | **78.6** | 89.2 | **94.4** | 64.4 | **99.7** | 78.2 | 83.1 | 82.2 | 80.5 | 92.9 |
| MAMUT-MathBERT-MLM | 70.6 | **99.6** | 82.7 | 88.4 | 77.3 | 87.8 | 93.9 | 58.9 | 98.9 | 73.8 | 85.1 | 85.2 | 81.6 | 93.8 |
| MAMUT-MPBERT-MLM | **78.3** | 99.4 | **87.6** | 87.2 | 78.1 | **89.4** | 94.2 | **67.7** | **99.7** | **80.6** | **87.3** | **86.5** | **84.6** | **94.8** |
| MAMUT-BERT | 97.2 | 100.0 | 98.6 | 96.5 | 83.3 | 96.7 | 97.5 | 96.9 | 99.9 | 98.4 | 99.7 | 99.3 | 99.0 | 99.7 |
| MAMUT-MathBERT | 97.5 | 100.0 | 98.7 | 97.0 | 83.6 | 97.2 | 97.8 | 95.2 | 99.9 | 97.5 | 97.7 | 98.6 | 98.5 | 99.4 |
| MAMUT-MPBERT | 98.1 | 100.0 | 99.1 | 97.0 | 83.7 | 97.4 | 97.8 | 97.9 | 100.0 | 98.9 | 99.7 | 99.6 | 99.4 | 99.8 |

Table 6: Results for NMF-FT and MFR-FT. All values are reported as percentages. Since the models MAMUT-BERT, MAMUT-MathBERT and MAMUT-MathBERT include the training data of the NMF and MFR tasks in their pretraining, they have a strong advantage and are therefore highlighted in gray. However, we note that these models did not see the test data of NMF or MFR as well, i.e., there is no information leak leading to this performance. Still, we focus our analysis of the comparison of fine-tuned models that only used MLM as pretraining task to separate the impact of the data from that of the pretraining tasks. **Bold** highlights the best result per metric excluding the fully pretrained models.

sity, ranging across topics, complexity levels, and reasoning steps, MAMUT complements these datasets by targeting a different aspect of variation: diversity in notation and symbols. This focus on symbolic diversity addresses a critical yet underexplored dimension in math-oriented NLP tasks. By systematically modifying the notation and symbols of formulas, MAMUT enforces a model's ability to generalize beyond simple patterns (e.g., the binomial formula contains $a$, $b$, and $c$), enabling better robustness in downstream tasks, as shown by our experiments. Moreover, due to the rule-based design, MAMUT tracks the transformations it applies, allowing for detailed, fine-grained evaluations (see Appendix C.4.1).

## 8.2 Data Quality and Reliability

Several measures are taken to ensure high data quality and reliability for all four generated datasets. For MF, we use only formulas judged generally valid by SymPy. MT is generated from high-quality textual inputs: the AMPS problems from Khan Academy, a reputable source of educational material, and the ARQMath dataset, which is based on community-moderated mathematical StackExchange discussions. To ensure quality, we include only ARQMath questions with at least five upvotes. The remaining datasets, NMF and MFR, are derived from the manually created NMFT of high-quality formulas. To avoid generating incorrect falsifications, care is taken to exclude transformations that preserve equivalence trivially (e.g., adding zero or multiplying by one). The symbolic transformations performed by EquVG, including parsing, symbol substitution, and randomized printing, are based on theoretically equivalence-preserving operations. Hence, the overall reliability of MAMUT-enhanced datasets is closely tied to the correctness of input formulas and the accuracy of SymPy's parsing, modification, and printing routines. Among the transformation steps, parsing seems to be the most error-prone due to the inherent complexity and ambiguity of mathematical notation. This issue is further discussed in the following section.

## 9 Limitations

While the experiments conducted in Section 7 demonstrate strong performance of MAMUT-generated datasets, several limitations remain that could be addressed in future work.

First, although the current implementation of MAMUT already supports a wide range of randomized notational variants, more equivalent transformations could be considered (e.g., $1 + 2 + \cdots + n$ vs. $\sum_{i=1}^{n} i$). As a

result, models trained on MAMUT-enhanced datasets may generalize well to covered notations but struggle with those not supported yet.

Second, the current version of MAMUT focuses exclusively on LaTeX-based input and output. While LaTeX is the most prevalent format for mathematical notation in science, the underlying SymPy framework supports additional formats for parsing (e.g., Mathematica and Maxima) and printing (e.g., MathML and various programming languages). This makes MAMUT easily extensible to other formats. Notably, the SymPy parsers require no changes for using them in MAMUT. Extending support to these formats would mainly require adapting the SymPy printers to handle randomized output decisions, while FalseVG and the symbol substitution logic in EquVG remains unchanged. Notably, this setup naturally enables translating formulas from one format into another while generating equivalent and falsified versions.

Third, while our generation process is entirely rule-based and includes safeguards to avoid incorrect falsifications, some edge cases may still result in undesired behavior. Moreover, undetected parsing errors (e.g., incompletely parsed formula) may be another problem. A comprehensive error analysis of MAMUT outputs, especially of FalseVG, would help to quantify the reliability of the generated datasets.

## 10 Conclusion

Mathematical formulas are essential to communicate complex and abstract concepts in various scientific fields. To effectively encode the unique structure of mathematical language, specialized mathematical language models are required. We developed MAMUT, a framework based on SymPy (Meurer et al., 2017) that generates equivalent and falsified versions of LaTeX formulas through parsing, substituting, possibly falsifying, and printing again into LaTeX format. MAMUT diversifies and expands datasets, as demonstrated by four generated large, high-quality datasets: MF, MT, NMF and MFR, all publicly available on Hugging Face (see Table 5). These datasets can be leveraged for further mathematical pretraining of language models utilizing tasks such as Masked Language Modeling (MLM) and Causal Language Modeling (CLM) to predict equation parts, a Next Sentence Prediction (NSP) variant that predicts if equations are equivalent, or contrastive learning between positive and negative samples to learn equation embeddings. Our experiments show that pretraining with these datasets consistently improves mathematical performance for BERT models, even for those with prior mathematical training.

### Acknowledgments

The authors gratefully acknowledge the computing time made available to them on the high-performance computer at the NHR Center of TU Dresden. This center is jointly supported by the Federal Ministry of Education and Research and the state governments participating in the NHR[1]. This paper is based on work conducted during J.D.'s Master's thesis at Dresden University of Technology. A.R. was a doctoral researcher at Dresden University of Technology during this time. A.R. was funded through the Azrieli international postdoctoral fellowship and the Ali Kaufman postdoctoral fellowship. We also thank Katja Noack for providing an initial version of the transformer pretraining code.

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

# A Original Datasets

In Table 7, we present one version of each mathematical identity of NMFT, while this entire raw dataset is available on Hugging Face[2] as part of the NMF dataset files. Subsequently, Table 8 provides an example entry of ARQMath (Mansouri et al., 2022a) from the Mathematical Stack Exchange, while Table 9 shows an example of AMPS (Hendrycks et al., 2021).

| Name | Formula |
|------|---------|
| Addition Theorem for Cosine | $\forall \alpha, \beta \in \mathbb{R} : \cos(\alpha + \beta) = \cos(\alpha)\cos(\beta) - \sin(\alpha)\sin(\beta)$ |
| Addition Theorem for Sine | $\forall \alpha, \beta \in \mathbb{R} : \sin(\alpha + \beta) = \sin(\alpha)\cos(\beta) + \cos(\alpha)\sin(\beta)$ |
| Addition Theorem for Tangent | $\forall \alpha, \beta \in \mathbb{R} : \tan(\alpha + \beta) = \frac{\tan(\alpha) + \tan(\beta)}{1 - \tan(\alpha)\tan(\beta)}$ |
| Alternating Harmonic Series | $\sum_{n=1}^{\infty} \frac{(-1)^{n+1}}{n} = 1 - \frac{1}{2} + \frac{1}{3} - \frac{1}{4} \pm \ldots = \ln(2)$ |
| Basel Problem | $\sum_{n=1}^{\infty} \frac{1}{n^2} = \frac{1}{1^2} + \frac{1}{2^2} + \frac{1}{3^2} + \frac{1}{4^2} + \frac{1}{5^2} + \frac{1}{6^2} + \ldots = \frac{\pi^2}{6}$ |
| Bayes' Theorem | $\mathbb{P}(A\|B) = \frac{\mathbb{P}(B\|A) \cdot \mathbb{P}(A)}{\mathbb{P}(B)}$ |
| Bernouilli Inequality | $\forall x \geq -1, \forall \alpha > 1 \Rightarrow (1 + x)^\alpha \geq 1$ |
| Binomial Coefficient Formula | $\forall n, k \in \mathbb{N}, n \geq k : \binom{n}{k} = \frac{n!}{k!(n-k)!}$ |
| Binomial Distribution | $\mathbb{P}(X = k) = \binom{n}{k} p^k \cdot (1 - p)^{n-k}$ |
| Binomial Series | $\forall \alpha, x \in \mathbb{C} > 0 : \|x\| < 1 \Rightarrow (1 + x)^\alpha = \sum_{k=0}^{\infty} \binom{\alpha}{k} x^k$ |
| Binomial Theorem | $\forall a, b \in \mathbb{R} \forall n \in \mathbb{N} : (a + b)^n = \sum_{k=0}^{n} \binom{n}{k} a^{n-k} b^k$ |
| Chain Rule | $\frac{d}{dx}[f(g(x))] = f'(g(x)) \cdot g'(x)$ |
| Complex Number Division | $\forall a, b, c, d \in \mathbb{R} : \frac{a+b\mathrm{i}}{c+d\mathrm{i}} = \frac{(ac+bd)+(bc-ad)i}{c^2+d^2}$ |
| Complex Number Inverse | $\forall z \in \mathbb{C} : z = a + b\mathrm{i} \Rightarrow z^{-1} = \frac{a}{a^2+b^2} - \frac{b}{a^2+b^2}\mathrm{i}$ |
| Complex Number Multiplication | $\forall a, b, c, d \in \mathbb{R} : (a + b\mathrm{i}) \cdot (c + d\mathrm{i}) = (ac - bd) + (ad + bc)\mathrm{i}$ |
| Complex Number Sum | $\forall a, b, c, d \in \mathbb{R} : (a + b\mathrm{i}) + (c + d\mathrm{i}) = (a + c) + (b + d)\mathrm{i}$ |
| Cosine Function Definition | $\forall x \in \mathbb{R} : \cos(x) = \sum_{n=0}^{\infty} \frac{(-1)^n}{(2n)!} x^{2n}$ |
| Covariance | $\mathrm{Cov}[X, Y] = \mathrm{E}[(X - \mathrm{E}[X])(Y - \mathrm{E}[Y])]$ |
| De Morgan Law | $\forall x, y : \neg(x \wedge y) = \neg x \vee \neg y$ |
| Derivative of Inverse Function | $\frac{d}{dx}[f^{-1}(x)] = \frac{1}{f'(f^{-1}(x))}$ |
| Derivative of a Function | $f'(x) = \lim_{h \to 0} \frac{f(x+h) - f(x)}{h}$ |
| Determinant of 2x2 Matrix | $\det \left( \begin{smallmatrix} a & b \\ c & e \end{smallmatrix} \right) = a \cdot e - b \cdot c$ |
| Determinant of 3x3 Matrix | $\det \left( \begin{smallmatrix} a & b & c \\ d & e & f \\ g & h & j \end{smallmatrix} \right) = a \cdot \det \left( \begin{smallmatrix} e & f \\ h & j \end{smallmatrix} \right) - b \cdot \det \left( \begin{smallmatrix} d & f \\ g & j \end{smallmatrix} \right) + c \cdot \det \left( \begin{smallmatrix} d & e \\ g & h \end{smallmatrix} \right)$ |
| Distributive Law of Sets | $A \cup (B \cap C) = (A \cup B) \cap (A \cup C)$ |
| Euler's Formula | $\forall \alpha \in \mathbb{C} : e^{\mathrm{i}\alpha} = \cos(\alpha) + \mathrm{i}\sin(\alpha)$ |
| Euler's Formula for Polyhedra | $V - E + F = 2$ |
| Euler's Identity | $e^{\mathrm{i}\pi} + 1 = 0$ |
| Euler's Number | $e = \lim_{n \to \infty} (1 + 1/n)^n$ |

Table 7: The 71 mathematical identities of the NMFT dataset.

---

[2] https://huggingface.co/datasets/ddrg/named_math_formulas/blob/main/data.json

(Continued from previous page)

| Name | Formula |
|------|---------|
| Expected Value | $\mathbb{E}(X) = \sum_{i=1}^{n} x_i \mathbb{P}(X = x_i)$ |
| Exponential Function | $\forall x \in \mathbb{R} : \lim_{n \to \infty} (1 + x/n)^n = e^x$ |
| Factorial | $\forall n \in \mathbb{N} : n! = 1 \cdot 2 \cdot 3 \cdot 4 \cdot \ldots \cdot n$ |
| First Binomial Formula | $\forall a, b \in \mathbb{R} : (a + b)^2 = a^2 + 2ab + b^2$ |
| Fundamental Theorem of Calculus | $\int_a^b f(x)\, dx = F(b) - F(a)$ |
| Gamma Function | $\forall n \in \mathbb{N} : \Gamma(n) = \int_0^\infty x^{n-1} e^{-x} dx = (n-1)!$ |
| Gaussian Integral | $\int_{-\infty}^{\infty} exp(-x^2) dx = \sqrt{\pi}$ |
| Geometric Series | $\sum_{n=0}^{\infty} r^n = \frac{1}{1-r}$ |
| Gregory-Leibniz Series | $\sum_{n=0}^{\infty} (-1)^n \cdot \frac{1}{2n+1} = \frac{\pi}{4}$ |
| Harmonic Series | $\sum_{n=1}^{\infty} \frac{1}{n} = \infty$ |
| Hölder Inequality | $\forall p, q > 1, \frac{1}{p} + \frac{1}{q} = 1, \forall x, y \in \mathbb{R}^n$ $\Rightarrow \sum_{i=1}^{n} |x_i y_i| \leq (\sum_{i=1}^{n} |x_i|^p)^{\frac{1}{p}} \cdot (\sum_{i=1}^{n} |y_i|^q)^{\frac{1}{q}}$ |
| Integration by Parts | $\int f(x) g'(x)\, dx = f(x) g(x) - \int g(x) f'(x)\, dx$ |
| Inverse of 2x2 Matrix | $\left( \begin{smallmatrix} a & b \\ c & d \end{smallmatrix} \right)^{-1} = \frac{1}{ad-bc} \left( \begin{smallmatrix} d & -b \\ -c & a \end{smallmatrix} \right)$ |
| Law of Cosines | $c^2 = a^2 + b^2 - 2ab \cos(C)$ |
| Law of Large Numbers | $\lim_{n \to \infty} \frac{1}{n} \sum_{i=1}^{n} x_i = [E](X)$ |
| Law of Sines | $\frac{\sin(A)}{a} = \frac{\sin(B)}{b} = \frac{\sin(C)}{c}$ |
| Law of Total Probability | $\mathbb{P}(A) = \sum_{i=1}^{n} \mathbb{P}(A|B_i) \mathbb{P}(B_i)$ |
| Logarithm Power Rule | $\forall b \in \mathbb{R}, b > 0, b \neq 1, \forall x, r \in \mathbb{R}, x > 0 : \log_b(x^r) = r \cdot \log_b(x)$ |
| Logarithm Product Rule | $\forall b \in \mathbb{R}, b > 0, b \neq 1, \forall x, y > 0 : \log_b(xy) = \log_b(x) + \log_b(y)$ |
| Logarithm Quotient Rule | $\forall b \in \mathbb{R}, b > 0, b \neq 1, \forall x, y > 0 : \log_b(x/y) = \log_b(x) - \log_b(y)$ |
| Minkowski Inequality | $\forall p > 1 \Rightarrow \sum_{i=1}^{n} |x_i + y_i|^{\frac{1}{p}} \leq (\sum_{i=1}^{n} |x_i|^p)^{\frac{1}{p}} + (\sum_{i=1}^{n} |y_i|^p)^{\frac{1}{p}}$ |
| Multiplication of 2x2 Matrix | $A = \left( \begin{smallmatrix} a & b \\ c & d \end{smallmatrix} \right), B = \left( \begin{smallmatrix} e & f \\ g & h \end{smallmatrix} \right) \Rightarrow A \cdot B = \left( \begin{smallmatrix} ae+bg & af+bh \\ ce+dg & cf+dh \end{smallmatrix} \right)$ |
| Normal Distribution | $f(x) = \frac{1}{\sigma \sqrt{2\pi}} e^{-(x-\mu)^2/(2\sigma^2)}$ |
| Pascal's Rule | $\forall n, k \in \mathbb{N} : \binom{n+1}{k+1} = \binom{n}{k+1} + \binom{n}{k}$ |
| Poisson Distribution | $\mathbb{P}(X = k) = \frac{e^{-\lambda} \lambda^k}{k!}$ |
| Power Rule | $\forall n \in \mathbb{R}, n \neq 0 : \frac{d}{dx}(x^n) = nx^{n-1}$ |
| Principle of Inclusion-Exclusion | $|A \cup B| = |A| + |B| - |A \cap B|$ |
| Product Rule | $\frac{d}{dx}[u(x) \cdot v(x)] = u'(x) \cdot v(x) + u(x) \cdot v'(x)$ |
| Pythagorean Identity | $\forall \alpha \in \mathbb{R} : \sin^2(\alpha) + \cos^2(\alpha) = 1$ |
| Pythagorean Theorem | $a^2 + b^2 = c^2$ |
| Quadratic Formula | $\forall a, b, c \in \mathbb{R}, a \neq 0 : a \cdot x^2 + b \cdot x + c = 0 \Rightarrow x_{1,2} = \frac{-b \pm \sqrt{b^2 - 4ac}}{2a}$ |
| Quotient Rule | $\forall b \in \mathbb{R}, b > 0, b \neq 1, \forall x, y > 0 : \log_b(x/y) = \log_b(x) - \log_b(y)$ |
| Riemann Zeta Function | $\forall z \in \mathbb{C}, \mathrm{Re}(z) > 1 : \zeta(z) = \sum_{n=1}^{\infty} \frac{1}{n^z}$ |

Table 7: The 71 mathematical identities of the NMFT dataset.

(Continued from previous page)

| Name | Formula |
|------|---------|
| Rule de l'Hôpital | $\lim_{x \to a} \frac{f(x)}{g(x)} = \lim_{x \to a} \frac{f'(x)}{g'(x)}$ |
| Second Binomial Formula | $\forall a, b \in \mathbb{R} : (a - b)^2 = a^2 - 2a \cdot b + b^2$ |
| Sine Function Definition | $\forall x \in \mathbb{R} : \sin(x) = \sum_{n=0}^{\infty} (-1)^n / (2n + 1)! x^{2n+1}$ |
| Stirling Approximation | $\forall n \in \mathbb{N} : n! \approx \sqrt{2\pi n} \left( \frac{n}{e} \right)^n$ |
| Taylor Series | $f(x) = \sum_{n=0}^{\infty} \frac{f^{(n)}(a)}{n!} (x - a)^n$ |
| Third Binomial Formula | $\forall a, b \in \mathbb{R} : (a + b)(a - b) = a^2 - b^2$ |
| Variance | $\mathbb{V}\mathrm{ar}[X] = \mathrm{E}\left[ (X - \mathrm{E}[X])^2 \right]$ |
| Wallis Product | $\prod_{n=1}^{\infty} \frac{4n^2}{4n^2 - 1} = \frac{\pi}{2}$ |
| Young Inequality | $\forall p, q > 1, 1/p + 1/q = 1, \forall a, b \geq 0 \Rightarrow ab \leq \frac{a^p}{p} + \frac{b^q}{q}$ |
| pq Formula | $\forall p, q \in \mathbb{R} : x^2 + px + q = 0 \Rightarrow x_{1,2} = -\frac{p}{2} \pm \sqrt{\frac{p^2}{4} - q}$ |

Table 7: The 71 mathematical identities used in NMFT.

# B MAMUT

Table 10 shows examples of the strategies for generating falsified formulas of FalseVG. Table 11 reports the used symbol groups for the symbol substitution of EquVG.

## B.1 Implementation

As discussed in Section 6, MAMUT relies on the EquVG and FalseVG algorithms, which generate equivalent or falsified versions of a given formula. We implemented these algorithms using the Python library SymPy 1.12, which is an open-source symbolic mathematics library with computer algebra system features (Meurer et al., 2017). This library includes a LaTeX parser for converting expressions into an internal SymPy representation, which can then be printed back into LaTeX. The SymPy formula representation is a symbolic expression, as required for EquVG and FalseVG, and supports the substitution of variables and generic functions. However, the built-in SymPy parser had limitations in handling various mathematical notations. The parsing capability has been expanded during this work, including the parsing of matrices, sets, derivatives, and various operators ($\pm$, $\cup$, $\cap$, $\mathbb{E}[X]$, $\mathbb{V}\mathrm{ar}[X]$, ...). Additionally, the SymPy LaTeX parsing was expanded to support a wider range of mathematical expressions through the implementation of an adaptive hybrid approach. This approach introduces a SymPy-like expression that enables safe string-based substitutions. As discussed earlier, a naive string replacement is inadequate for mathematical symbol substitution. For instance, if we replace x in \exp{x}, it would also unintentionally replace the occurrence of x within \exp. To address this issue, our implementation of the safe string-based substitution detects such situations resulting in a failure to avoid invalid expressions during the generation of versions ensuring high data quality. This SymPy-like expression also utilizes a predefined list of known symbols and LaTeX commands that, when present in the input, are excluded from substituting. For example, the \exp command is included in this list, allowing \exp{x} to be substituted using our safe string-based approach. This method, while being less powerful than the classical SymPy expressions, extends substitution support to a wide range of mathematical notations that can not be parsed in the classical parser. Hence, the hybrid combination of the classical SymPy expression with randomized printing and the string-based substitution, supporting a wider range of operators, aligns perfectly with our need to create a diverse, high-quality mathematical dataset with substituted symbols.

| Title | Derivative of sigmoid function $\sigma(x) = \frac{1}{1+e^{-x}}$ |
|---|---|
| Question | In my AI textbook there is this paragraph, without any explanation. The sigmoid function is defined as follows: "$\sigma(x) = \frac{1}{1+e^{-x}}$. This function is easy to differentiate because $\frac{d\sigma(x)}{d(x)} = \sigma(x) \cdot (1 - \sigma(x))$." It has been a long time since I've taken differential equations, so could anyone tell me how they got from the first equation to the second? |
| Answer 1 | Consider $f(x) = \frac{1}{\sigma(x)} = 1 + e^{-x}$ . Then, on the one hand, the chain rule gives $f'(x) = \frac{d}{dx}\left(\frac{1}{\sigma(x)}\right) = -\frac{\sigma'(x)}{\sigma(x)^2}$, and on the other hand, $f'(x) = \frac{d}{dx}\left(1 + e^{-x}\right) = -e^{-x} = 1 - f(x) = 1 - \frac{1}{\sigma(x)} = \frac{\sigma(x)-1}{\sigma(x)}$. Equate the two expressions, and voilà! |
| Answer 2 | Let's denote the sigmoid function as $\sigma(x) = \dfrac{1}{1 + e^{-x}}$. The derivative of the sigmoid is $\dfrac{d}{dx}\sigma(x) = \sigma(x)(1 - \sigma(x))$ Here's a detailed derivation: $$\begin{aligned} \frac{d}{dx}\sigma(x) &= \frac{d}{dx}\left[\frac{1}{1+e^{-x}}\right] \\ &= \frac{d}{dx}\left(1+e^{-x}\right)^{-1} \\ &= -(1+e^{-x})^{-2}(-e^{-x}) \\ &= \frac{e^{-x}}{(1+e^{-x})^2} \\ &= \frac{1}{1+e^{-x}} \cdot \frac{e^{-x}}{1+e^{-x}} \\ &= \frac{1}{1+e^{-x}} \cdot \frac{(1+e^{-x})-1}{1+e^{-x}} \\ &= \frac{1}{1+e^{-x}} \cdot \left(\frac{1+e^{-x}}{1+e^{-x}} - \frac{1}{1+e^{-x}}\right) \\ &= \frac{1}{1+e^{-x}} \cdot \left(1 - \frac{1}{1+e^{-x}}\right) \\ &= \sigma(x) \cdot (1 - \sigma(x)) \end{aligned}$$ |

Table 8: Example entry of the ARQMath dataset with preserved LaTeX formatting (post ID 78575, answer IDs 78578 and 1225116).

| Problem | Simplify the following expression: $y = \dfrac{p^2 - 3p - 54}{p - 9}$ |
|---|---|
| Answer/ Hints | First factor the polynomial in the numerator. $p^2 - 3p - 54 = (p-9)(p+6)$. So we can rewrite the expression as: $y = \dfrac{(p - 9)(p + 6)}{p - 9}$. We can divide the numerator and denominator by $(p - 9)$ on condition that $p \neq 9$. Therefore $y = p + 6; p \neq 9$. |

Table 9: Example entry of the AMPS dataset (file `amps/khan/504/1607900679.json`).

| | Original Formula | Falsified Formula | Description |
|---|---|---|---|
| **⚌ Equality** | $a^2 + b^2 = c^2$ | $a^2 + b^2 = c^2 - 1$ | Subtracted 1 from right side |
| | | $a^2 = c^2$ | Removed $b^2$ |
| | | $a^2 + b^{2+x} = c^2$ | Inserted $+x$ in exponent of $b^2$ |
| **≤ Inequality** | $x > y$ | $x \leq y$ | Inverted $>$ to $\leq$ |
| | $ab \leq \dfrac{a^2 + b^2}{2}$ | $ab > \dfrac{a^2 + b^2}{2}$ | Inverted $\leq$ to $>$ |
| | $x \neq 0$ | $x = 0$ | Inverted $\neq$ to $=$ |
| **⤮ Swap** | $a^2 + b^2 = c^2$ | $a^2 + 2^b = c^2$ | Swapped $b$ and 2 in $b^2$ |
| | $F(a) - F(b)$ | $F(b) - F(a)$ | Swapped order of arguments |
| | $\ln\left(\dfrac{x}{y}\right) = \ln(x) - \ln(y)$ | $\ln\left(\dfrac{x}{y}\right) = \sin(x) - \ln(y)$ | Replaced ln by sin in $\ln(x)$ |
| | $\dfrac{sin(\alpha)}{a} = \dfrac{\sin(\beta)}{b}$ | $\dfrac{\log(\alpha)}{a} = \dfrac{\sin(\beta)}{b}$ | Replaced sin by log in $\sin(\alpha)$ |
| **A Variable** | $n! = 1 \cdot 2 \cdot \ldots \cdot n$ | $k! = 1 \cdot 2 \cdot \ldots \cdot n$ | Replaced $n$ by $k$ in $n!$ |
| | $\displaystyle\sum_{i=1}^{n} i^2$ | $\displaystyle\sum_{i=1}^{n} k^2$ | Replaced $i$ by $k$ only in $i^2$ |
| **∞ Constant** | $e^{i\pi} = -1$ | $3^{i\pi} = -1$ | Replaced $e$ by 3 |
| | | $e^{1\pi} = -1$ | Replaced i by 1 |
| | | $e^{ie} = -1$ | Replaced $\pi$ by $e$ |
| | | $42^{i\pi} = -1$ | Replaced $e$ by 42 |
| | $\displaystyle\sum_{i=1}^{\infty} \dfrac{1}{i^2} = \dfrac{\pi^2}{6}$ | $\displaystyle\sum_{i=1}^{n} \dfrac{1}{i^2} = \dfrac{\pi^2}{6}$ | Replaced $\infty$ by $n$ |
| **⫟ Distribute** | $\sin(x) + \sin(y)$ | $\sin(x + y)$ | Applied sine additivity |
| | $\dbinom{n}{k} = \dfrac{n!}{k!(n-k)!}$ | $\dbinom{n}{k} = \dfrac{n!}{(k \cdot (n-k))!}$ | Applied faculty multiplicity |
| | | $\dbinom{n}{k} = \dfrac{n!}{k! \cdot (n! - k!)}$ | Applied faculty multiplicity |
| **👥⚙ Manual** | $\forall n \in \mathbb{N} : n! = \ldots$ | $\forall n \in \mathbb{R} : n! = \ldots$ | Replaced $\mathbb{N}$ by $\mathbb{R}$ |
| | $a^2 + b^2 = c^2$ | $a^2 = b^2 + c^2 - 2bc\cos(\alpha)$ | Similar formula |
| | In any right-angled triangle ... | In any right-angled square ... | Replaced "triangle" by "square" |
| **🎲 Random** | $a^2 + b^2 = c^2$ | $\sin^2(\alpha) + \cos^2(\alpha) = 1$ | Random formula |
| | In any right-angled triangle ... | The derivative of a function $f$ is ... | Random text |

Table 10: Examples of the strategies for generating falsified formulas (FalseVG).

| | Symbol Groups | Typical Context | Example |
|---|---|---|---|
| **Variables** | $a, b, c, d, e, f, g, h$ | Parameters | $ax^2 + bx + c = 0$ |
| | $i, j, k, l$ | Indices | $C_{ij} = \sum_k A_{ik} B_{kj}$ |
| | $k, l, m, n$ | Counts | $\binom{n}{k} = \frac{n!}{k!(n-k)!}$ |
| | $p, q, r, s, t$ | Parameters, Points | $x^2 + px + q = 0$ |
| | $u, v, w$ | Vectors | $u \times v = w$ |
| | $x, y, z$ | Unknowns | $x + 2y + 3z = 4$ |
| | $A, B, C, D, E, F, G, H$ | Matrices, Sets | $A \cup (B \cap C) = (A \cup B) \cap (A \cup C)$ |
| | $Q, R, S, T, U, V, W, X, Y, Z$ | Random Variables | $X = Y - Z$ |
| | $\alpha, \beta, \gamma, \delta, \theta, \vartheta, \psi, \phi, \varphi, \rho$ | Angles | $\alpha + \beta + \gamma = 180°$ |
| | $\tau, \sigma, \lambda, \mu, \nu$ | Scalars | $\lambda \left( \begin{smallmatrix} x_1 \\ x_2 \end{smallmatrix} \right) + \mu \left( \begin{smallmatrix} y_1 \\ y_2 \end{smallmatrix} \right) = 0$ |
| **Functions** | $f, g, h, u, v$ | Generic Functions | $[uv]' = u'v + uv'$ |
| | $F, G, H, U, V$ | Antiderivatives | $\int_a^b f(x)dx = F(b) - F(a)$ |
| | $\tau, \sigma, \lambda, \mu, \nu$ | Permutations | $\sigma \circ (\tau \circ \mu) = (\sigma \circ \tau) \circ \mu$ |

Table 11: Defined symbol groups for the symbol substitution of MAMUT.

In addition, our SymPy parser implementation is adaptive. Even if an input formula can not be parsed classically, the classical parsing still succeeds for parts of it. As a result, formulas are split at delimiter symbols such as : or `\Rightarrow`. Using these extended parsing capabilities, the input `\forall x, y: x\cdot y=y\cdot x` is parsed into two sub-expressions: `\forall x, y`, which can not be parsed classically with the used implementation, and `x\cdot y=y\cdot x`, which is parsed into a classical SymPy expression. Both sub-expressions support the substitution of x and y. This results in, for instance, `\forall \alpha, a: \alpha\times a=a\times \alpha`, where randomized printing was incorporated for the right subexpression. Similarly, support for parsing entire texts containing formulas enclosed within dollar symbols, denoting the LaTeX mathematical inline mode, is integrated into the SymPy parser. To create randomized LaTeX formulas from the parsed SymPy expressions, the SymPy LaTeX printer has been enhanced to support randomized decisions. The printing process is guided by randomized settings[3], which define all the randomized decisions the printer could make. The modified SymPy code is accessible in a forked repository on GitHub[4], providing a simple interface for generating equivalent and falsified versions of a formula. Additionally, the generation code for the generated datasets based on AMPS, ARQMath, and NMFT is publicly available[5], including the logic for base formula filtering, extraction, and validation. We filter the input formulas considered for MF using two SymPy methods. A formula is retained if either `sympy.solve()` finds at least one solution (e.g., $x = \pm\sqrt{2}$ for $x^2 = 2$), which can then be used as an implication (e.g., $x^2 = 2 \Rightarrow x = -\sqrt{2}$ or $x = \sqrt{2}$), or `sympy.simplify()` evaluates as `True` (e.g., $1 + 2 = 3$ and $\tan(x) = \frac{\sin(x)}{\cos(x)}$).

## B.2 Generated Datasets

To illustrate the behavior of MAMUT and the data extraction process, we provide artificial examples for each generated dataset based on the previously shown raw data: MF in Table 12, MT in Table 13, NMF in Table 14, and MFR in Table 15. Please note that not all examples are verified as part of the actual generated datasets, but are selected to illustrate the diversity of MAMUT.

---

[3]https://github.com/aieng-lab/sympy-random-LaTeX/blob/master/sympy/settings.py
[4]https://github.com/aieng-lab/sympy-random-LaTeX
[5]https://github.com/aieng-lab/math-mutator

| Formula |
| --- |
| $1 - \frac{1}{\sigma(x)} = \frac{\sigma(x)-1}{\sigma(x)}$ |
| $-\dfrac{1}{\tau(y)} + 1 = \frac{1}{\tau(y)} \cdot (-1 + \tau(y))$ |
| $\frac{1}{\nu(x)} * (\nu(x) + (-1)) = 1 - 1/\nu(x)$ |
| $(\lambda(x) + (-1))/\lambda(x) = 1 - 1/\lambda(x)$ |
| $1 - 1/\nu(x) = ((-1) + \nu(x))/\nu(x)$ |
| $-1/\mu(x) + 1 = \frac{1}{\mu(x)} \cdot (\mu(x) + (-1))$ |
| $\lambda(x) + (-1))/\lambda(x) = 1 - \frac{1}{\lambda(x)}$ |
| $p^2 - 3p - 54 = (p - 9)(p + 6)$ |
| $(p + 9 \cdot (-1)) \cdot (6 + p) = p^2 - p \cdot 3 - 54$ |
| $p^2 - 3 \cdot p + 54 \cdot (-1) = (9 \cdot (-1) + p) \cdot (p + 6)$ |
| $(p + 6) \cdot (-9 + p) = 54 \cdot (-1) + p^2 - p \cdot 3$ |
| $(p - 9)(p + 6) = p * p - p * 3 + 54(-1)$ |

Table 12: Example entries for MF (based on Table 8 and Table 9).

| Text |
| --- |
| In my AI textbook there is this paragraph, without any explanation. The sigmoid function is defined as follows: "$\sigma(x) = \frac{1}{1+e^{-x}}$. This function is easy to differentiate because $\frac{d\sigma(x)}{d(x)} = \sigma(x) \cdot (1 - \sigma(x))$." It has been a long time since I've taken differential equations, so could anyone tell me how they got from the first equation to the second? |
| In my AI textbook there is this paragraph, without any explanation. The sigmoid function is defined as follows: "$\tau(y) = 1/(e^{-y} + 1)$. This function is easy to differentiate because $\tau(y)(-\tau(y) + 1) = \tau'(y)$." It has been a long time since I've taken differential equations, so could anyone tell me how they got from the first equation to the second? |
| Consider $f(x) = \dfrac{1}{\sigma(x)} = 1 + e^{-x}$. Then, on the one hand, the chain rule gives $f'(x) = \frac{d}{dx}\left(\frac{1}{\sigma(x)}\right) = -\frac{\sigma'(x)}{\sigma(x)^2}$, and on the other hand, $f'(x) = \frac{d}{dx}(1 + e^{-x}) = -e^{-x} = 1 - f(x) = 1 - \frac{1}{\sigma(x)} = \frac{\sigma(x)-1}{\sigma(x)}$. Equate the two expressions, and voilà! |
| Consider $u(y) = 1/\sigma(y) = 1 + e^{-y}$. Then, on the one hand, the chain rule gives $\frac{d}{dy}u(y) = \frac{d}{dy}\frac{1}{\sigma(y)} = -\frac{1}{\sigma^2(y)}\frac{d}{dy}\sigma(y)$, and on the other hand, $u'(y) = \frac{d}{dx}(1 + e^{-y}) = -e^{-y} = 1 - u(y) = 1 - \frac{1}{\sigma(y)} = \frac{\sigma(y)-1}{\sigma(y)}$. Equate the two expressions, and voilà! |
| Simplify the following expression: $y = \dfrac{p^2 - 3p - 54}{p - 9}$ First factor the polynomial in the numerator. $p^2 - 3p - 54 = (p - 9)(p + 6)$. So we can rewrite the expression as: $y = \dfrac{(p - 9)(p + 6)}{p - 9}$. We can divide the numerator and denominator by $(p - 9)$ on condition that $p \neq 9$. Therefore $y = p + 6; p \neq 9$. |
| Simplify the following expression: $\frac{-54+s^2-s\cdot3}{s-9} = z$ First factor the polynomial in the numerator. $s * s - 3 * s - 54 = (s - 9) * (6 + s)$. So we can rewrite the expression as: $z = \frac{1}{s-9} \times (s - 9) \times (6 + s)$. We can divide the numerator and denominator by $-9 + s$ on condition that $s \neq 9$. Therefore $z = s + 6; s \neq 9$. |

Table 13: Example entries of MT (based on Table 8 and Table 9).

| Name | Formula | Label |
|------|---------|-------|
| Factorial | $d! = 1 \cdot 2 \cdot 3 \cdot 4 \cdot 5 \cdot \ldots \cdot d$ | ✓ |
| Definition of a factorial | $\forall n \in \mathbb{N} : n! = \prod_{i=1}^{\xi} i$ | ✗ |
| Definition of a factorial | $\forall n \in \mathbb{N} : (n+1)! = (n+n) \cdot n! \wedge 0! = 1$ | ✗ |
| Definition of a factorial | For any natural number $k$ we have $k!$ is defined as $k \coloneqq \prod_{j=1}^{k} j$. | ✗ |
| Factorial | For any natural number $n$, $n!$ can be obtained by multiplying all natural numbers from 1 to $Y$ together. | ✗ |
| Definition of a factorial | $\forall n, j \in \mathbb{N}, n \geq j : \binom{n}{j} = \frac{1}{j! \cdot (n-j)!} \cdot n!$ | ✗ |
| Factorial | $1 \cdot 2 \cdot 3 \cdot \frac{1}{4} \ldots n = n!$ | ✗ |
| Factorial | $\forall m \geq 1 : m! = m \cdot (m + (-1))!, 0! = 0$ | ✗ |
| Definition of a factorial | $1 * 2 * 3 * 4 \ldots x = x!$ | ✓ |
| Definition of a factorial | $k! = (1 - 3) \cdot 18 \cdot 4 \cdot 5 / \cdots \cdot n$ | ✗ |
| Factorial | $n! = \sum_{i=1}^{n} i$ | ✗ |
| Factorial | The sum of two complex numbers $g_1 + \mathrm{i} \cdot h = z$ and $g_2 + \mathrm{i} \cdot f = w$ is defined as $g_1 + g_2 + i * (h + f) = w + z$. | ✗ |
| Definition of a factorial | $\theta! = 1 \cdot 2 \cdot \ldots \cdot \theta$ | ✓ |

Table 14: Example entries of NMF (based on Table 4).

| Formula 1 | Formula 2 | Label |
|-----------|-----------|-------|
| The value of $(1+1/\tau)^\tau$ approaches the constant $e$ as $\tau$ tends to infinity. | As $\mu$ approaches infinity, the expression $(1 + 1/\mu)^\mu$ converges to the value of $e \approx 2.718$. | ✓ |
| By utilizing the infinite series $\sum_{n=0}^{\infty} z^{1+2n} \frac{(-1)^n}{(1+2n)!}$, we can define $\sin(z)$ for all real numbers $z$. | For all real numbers $x$ the expression $\sin(z)$ is defined as the infinite sum $\sum_{n=0}^{\infty} x^{2 \cdot n + 1} \cdot (2 \cdot n + 1)! \cdot (-1)^{-n}$. | ✗ |
| The limit as $l$ approaches infinity of the expression $\left(1 + \frac{1}{l} \cdot y\right)^l$ converges to the exponential function $e^y$ for any real number $y$. | $\forall x \in \mathbb{C} : e^x = \sum_{k=0}^{\infty} -k^x/k! = 1 + x + x^2/2! + x * x^2/3! + \ldots$ | ✗ |
| For all real positive $g$ with $g \neq 1$ and for all real positive $s, y$, we have $\log_b(sy) = \log_b(s) + \log_b(y)$. | For all real bases $b$ such that $0 < b$ and $b \neq 1$ and for all real positive $z, y$, the equality $\log_b(z/y) = \log_b(z) - \log_b(y)$ holds. | ✗ |
| The derivative of a composite function $f(g(z))$ with respect to $z$ is given by $\frac{d}{dg(z)} f(g(z)) \cdot \frac{d}{dz} g(z)$. | The derivative of a composite function $f(g(y))$ with respect to $y$ is given by $\frac{\mathrm{d}}{\mathrm{d}g(u)} f(g(u))/(\frac{\mathrm{d}}{\mathrm{d}u} g(u))$. | ✗ |
| $\forall m \geq 1 : m! = m \cdot (m + (-1))!, 0! = 1$ | $\forall a \in \mathbb{N} : (a+1)! = (a+1) \cdot a!, 0! = 1$ | ✓ |
| Let $c$ and $b$ be real numbers. In this case, $(c+b)(-b+c)$ is equal to $c^2 - b^2$. | $\frac{1}{b-b}(a+b) = -b^2 + a^1$ | ✗ |

Table 15: Example entries of MFR.

| Model | Hugging Face Identifier | Reference |
|-------|------------------------|-----------|
| BERT | `bert-base-cased` | Devlin et al. (2019) |
| MathBERT | `tbs17/MathBERT` | Shen et al. (2021) |
| MPBERT | `AnReu/math_pretrained_bert` | Reusch et al. (2022) |

Table 16: Baseline models used for comparison in our experiments.

### B.3 Analysis of NMF

For a better understanding of the version generation algorithms, EquVG and FalseVG, we delve into a more detailed analysis of the generated NMF dataset, visualized in Figure 5.

Figure 5a shows the distribution of strategies over the mathematical identities of the NMF dataset. The figure illustrates the proportions of how many falsified versions of a mathematical identity utilized a particular strategy. Since multiple strategies might be applied to generate a single falsified version, the proportions do not sum up to 100% per identity. Approximately half of the time, only a single strategy is applied. In general, the different strategies obviously have different proportions across the mathematical identities. The most common strategies are Variable and Swap because variables and swappable expressions (e.g., $\sin(x + y) \to \sin(x) + \sin(y)$) occur in almost all formulas, often even multiple times. About 20% of the strategies are intentionally completely random to avoid introducing a bias towards challenging negative examples. In real-world MIR applications, random pairs are more common than challenging ones. Some strategies can not be applied to certain identities, particularly the strategy inequality, which is not applicable to most identities. The reason why some identities containing no inequalities still have a nonzero proportion for the strategy Inequality is that this strategy can be applied even after a random or manual formula (containing an inequality) is chosen as a falsified version.

Another analysis can be deducted from Figure 5b, which shows the distribution of whether a variable, function or any of them has been replaced in a generated version of a mathematical identity in the NMF dataset. Since only ten identities contain generic function symbols, substitutions of functions can only be applied to those identities regularly. Again, we recognize proportions slightly above zero for many identities not containing generic functions due to function substitutions after applying the strategies Random or Manual. The overall proportion of substituted formulas is 52.3%, but when considering only equivalent versions, this proportion rises to 81.3%. For falsified versions, the substitution proportion is about 45.1%.

## C Experiments

This section provides implementation details and further results for the experiments conducted in Section 7. The model checkpoints used for comparison and starting point for further mathematical pretraining are reported in Table 16. Note that we also tried the MathBERT variant with custom mathematical vocabulary (`tbs17/MathBERT-custom`), but found that it performs weaker than MathBERT, and therefore excluded it from this study.

### C.1 Pretraining

This section provides details regarding the mathematical pretraining tasks and the implementation. We publish our code[6] for mathematical pretraining, and our fully trained mathematical models (see Table 17).

### C.1.1 Tasks

In this section, we introduce two domain-adapted MLM and two domain-adapted NSP tasks used for the training of mathematical BERT models based on MAMUT-enhanced data. MLM and NSP have proven

---

[6]https://github.com/aieng-lab/transformer-math-pretraining

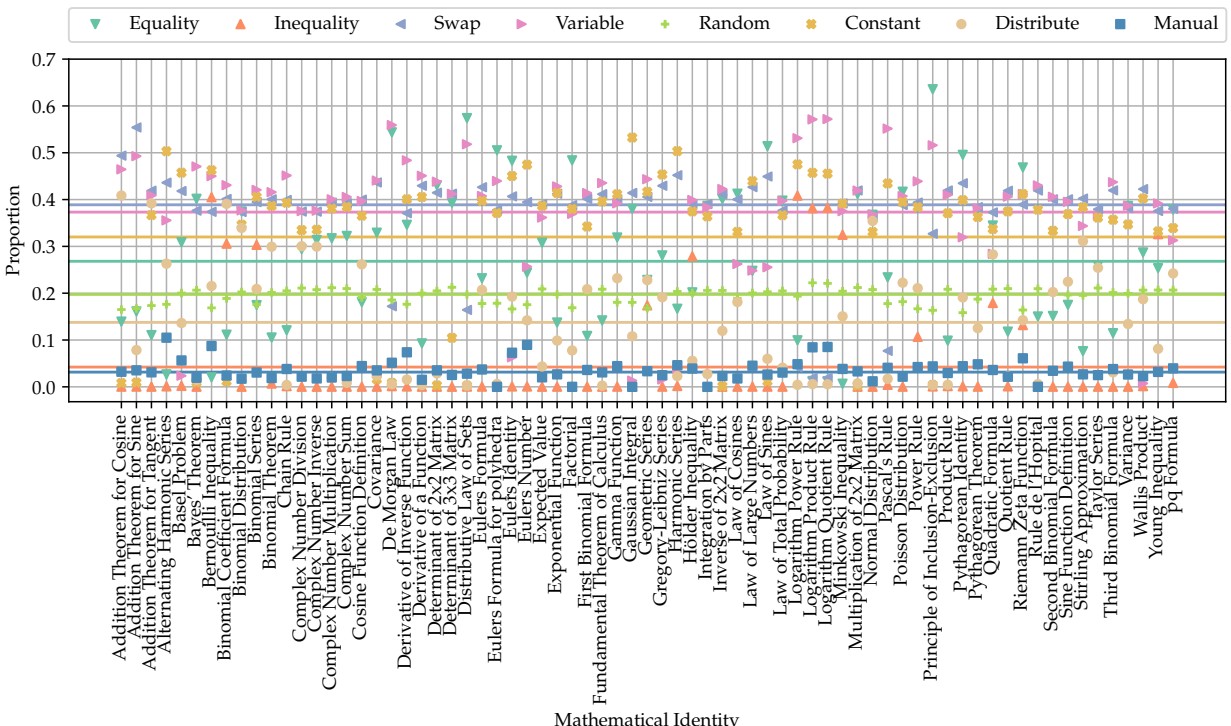

(a) Proportions of strategies used for generating falsified versions in NMF dataset per mathematical identity.

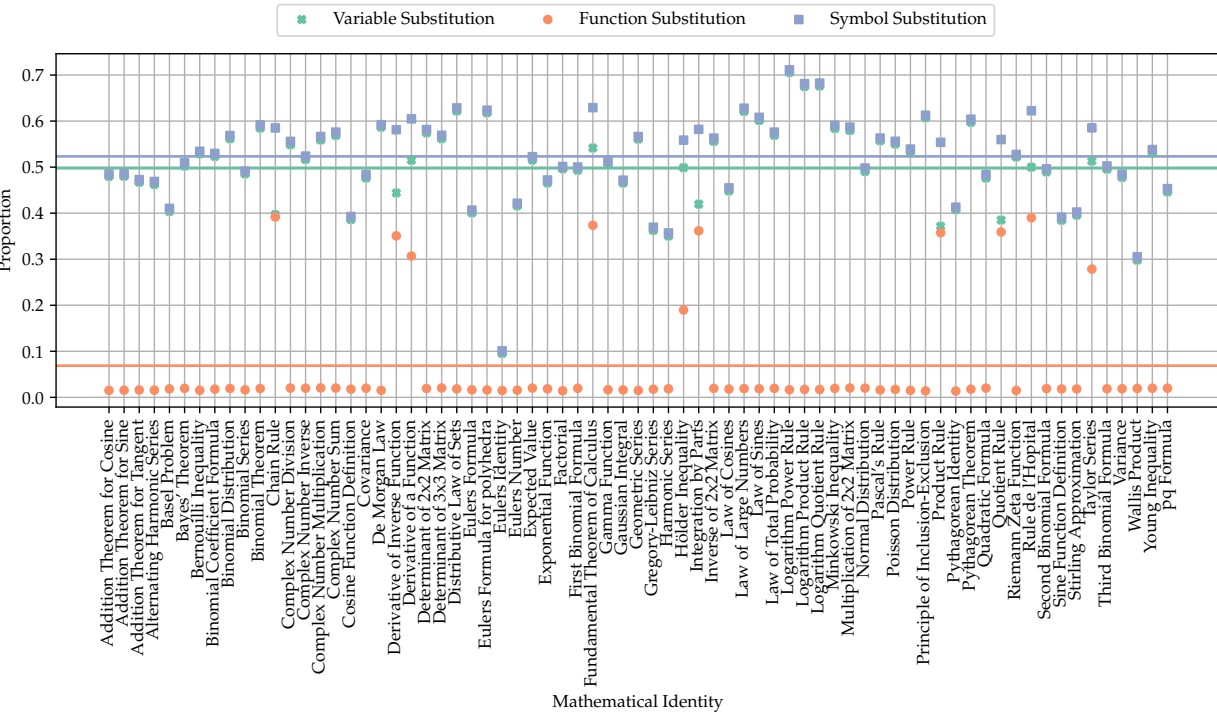

(b) Proportion of generated (equivalent and falsified) versions with at least one variable, function, or symbol substitution (variable or function).

Figure 5: Analysis of NMF. The mean across all identities is printed as a solid line.

| Model | Hugging Face Identifier | Source Model |
|-------|------------------------|--------------|
| MAMUT-BERT | `aieng-lab/bert-base-cased-mamut` | `bert-base-cased` |
| MAMUT-MathBERT | `ddrg/MathBERT-mamut` | `tbs17/MathBERT` |
| MAMUT-MPBERT | `ddrg/math_pretrained_bert_mamut` | `AnReu/math_pretrained_bert` |

Table 17: Hugging Face identifiers of our published mathematical models.

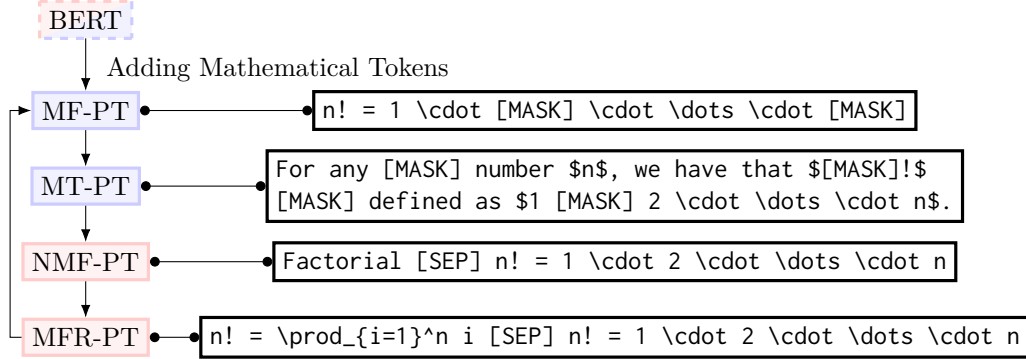

Figure 6: Visualization of the additional mathematical pretraining with all pretraining tasks (i.e., the training of MAMUT-BERT). MLM-like pretraining tasks are colored in blue, NSP-like in red.

to be effective in literature (Devlin et al., 2019; Feng et al., 2020; Reusch et al., 2022). Each of the four MAMUT-generated datasets (MF, MT, NMF, and MFR) create its dedicated pretraining task (denoted by *<dataset>-PT*), see Figure 6 for an overview. In the following, we briefly discuss the mathematical peculiarities.

**MF-PT** This pretraining task adapts the standard MLM by utilizing mathematical formulas instead of natural language text. It applies the masked language modeling, where 15% of the tokens are masked following the proposed setting of Devlin et al. (2019) for BERT: 80% of selected tokens are replaced with the special `[MASK]` token, 10% are replaced with a random token, and 10% remain unchanged. This task intends to train the model basic mathematical knowledge and understanding. Importantly, the formulas in MF have been carefully filtered such that only generally valid formulas are contained, which ensures that a masked token can be logically inferred by context (see Section 6 for details).

**MT-PT** While MF-PT is designed to learn intra-formula dependencies, MT-PT focuses on understanding relationships between formulas and natural language. That is why this pretraining task uses mathematical texts as input. To enforce masking of the mathematical parts in the text, a special masking algorithm is applied, which involves two additional masking probabilities:

- *mlm-formula-probability*: This probability determines the proportion of formula tokens being masked (i.e., tokens inside dollar tokens indicating the mathematical LATEX inline mode). We set *mlm-formula-probability* to 20%, meaning 20% of all formula tokens are masked.

- *mlm-math-words-probability*: This probability controls the masking of mathematical words, such as *sum*, *minus*, *equals*, *less*, or *function*. We collected 219 of such words manually[7]. We use *mlm-math-words-probability* = 30%, meaning about 30% of the tokens belonging to mathematical words within the sequence are masked. These words are either masked completely or not at all, effectively applying a whole word masking (Cui et al., 2021) to these mathematical words.

---

[7]https://github.com/aieng-lab/transformer-math-pretraining/blob/main/src/pretraining_methods/mlm_like/MLM/math_words.py

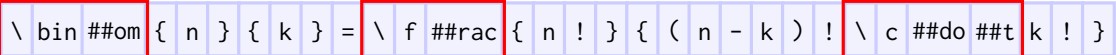

Figure 7: Example for different tokenization with the original BERT base tokenizer and their enriched versions. Red boxes mark split up tokens in the original tokenizer, which are tokenized as a single token in the enriched mathematical tokenizer.

If the total number of masked tokens is below 15%, additional tokens are selected at random to ensure that 15% of tokens are masked in total.

**NMF-PT** This NSP-like task associates a name of a mathematical identity with its formula or a describing text. Since for each positive sample four times more negative examples are generated (see Section 6), an unseen negative example can be used every epoch (as we train less than four epochs). Thus, within each epoch, a balanced number of positive and negative pairs are provided. The positive pairs are the same over the epochs, only the negative pairs change between the epochs.

**MFR-PT** Similar to the previous task, two separated sequences are used as model input in a NSP-like style. However, in this task, two formulas are used instead of a name and a formula, similar to the the second task of the ARQMath competition, where a formula is used as a query to find similar formulas (Mansouri et al., 2022a). This data is generated from NMF, by replacing the formula name with an equivalent version of the formula (from another positive sample of the mathematical identity within the dataset). This approach preserves the proportion between positive and negative pairs, and the negative samples are split over the epochs as for NMF-PT.

### C.1.2 Pretraining Setup

We combine multiple of these four mathematical pretraining tasks to further pretrain existing BERT based models. A visualization of such a mathematical pretraining with all four tasks is shown in Figure 6, including illustrative example samples. We apply our pretraining tasks not consecutively (i.e., one task is fully completed before starting the next one), but in a mixed manner, where the tasks alternate after each batch. For example, when training four tasks in this style on 8 GPUs, there are two dedicated GPUs for each task. Experiments show that the mixed training style enhances the overall training results, as the model does not forget previously learned information. Each task is trained for 250k optimization steps.

To enhance mathematical encodings, mathematics-specific tokens are added to the BERT model, whose randomly initialized embeddings are learned during the mathematical pretraining. This approach is similar as seen by other mathematical models (Shen et al., 2021; Reusch et al., 2022; Novotný & Štefánik, 2022). Before further mathematically pretraining, we enriched the vocabularies of BERT with the 300 most common LaTeX tokens found in MT that were not originally part of the tokenizer vocabulary. We defined LaTeX tokens as either a word starting with a backward slash (e.g., \frac) or a LaTeX environment identifier, i.e., text inside \begin{...} and \end{...} (e.g., pmatrix). The influence of the enriched BERT tokenizer regarding the tokenization of an example formula is shown in Figure 7. The word embeddings for these 300 newly added LaTeX tokens were initialized randomly and learned during the mathematical pretraining. We use the same initial weights across all mathematical pretrained BERT models. The token enrichment should help the model to better understand the word embeddings of frequently occurring terms in the context of mathematical LaTeX texts. Note that while this enrichment helps in capturing the correct semantic structure of mathematical content more effectively (Peng et al., 2021), the tokenization may still not always perfectly capture mathematical semantics. For instance, in both base models, the tokenization of the formula ab, representing multiplication of a and b with omitted multiplication symbol, results in a single token. Notice that these mathematical tokens have been only added with BERT as the starting model for mathematical pretraining. MPBERT already added 501 mathematical tokens as part of its mathematical pretraining (Reusch et al., 2022). MathBERT is available in two variants, one using the standard BERT vocabulary and another with a customized vocabulary (Shen et al., 2021). However, the variant without special tokens

performed better on our downstream tasks. Therefore, we used the version with the standard vocabulary, without any vocabulary enrichment.

### C.1.3 Implementation Details

For all our experiments, we employed programs written in Python 3.10. and utilize the libraries transformers 4.25.1 (Wolf et al., 2020) and datasets 2.10.1 from Hugging Face for tokenization and model training. For pretraining, a custom implementation of a training loop has been used relying on the PyTorch library (torch 1.13.1+cu117) to enable parallelized training on multiple GPUs. We initialize a separate new prediction head for each specific objective, following a similar approach to the implementation of the transformers library. All of our models introduced in the next section were pretrained on eight A100 GPUs. For all pretrained models, we maintained a batch size of 16 per GPU, 200 warm-up steps, a maximum input length of 512 tokens, a learning rate of $2 \times 10^{-5}$, and trained for four epochs. Training one model took about 12 hours per applied task.

### C.2 Fine-Tuning

We reuse NMF-PT and MFR-PT with a revised interpretation. During pretraining, the model's objective is to determine whether a given input, a name or a formula, can be associated with a formula, essentially creating a binary classification task. In the downstream task, we adapt this objective to use a name or formula as a query to retrieve relevant formulas as a ranked MIR task. Despite this reinterpretation, the training process remains the same, although only a reduced version of the dataset is used compared to pretraining. We denote these tasks as NMF-FT and MFR-FT, respectively.

To apply these downstream tasks, we created reduced versions of the datasets NMF and MFR that were used for pretraining. For each mathematical identity, we keep 250 positive examples and ten times more negative examples. Importantly, we maintain the train-test split of the pretraining, ensuring that samples in the downstream task's test set have never been seen during training. Importantly, the pretraining test data has *not* influenced any training decisions, such as early stopping, model selection, or any other procedure that could introduce information leakage. Specifically, the pretraining test set was further divided into two splits: one for validation and one for testing. All splits are sampled in a stratified manner to preserve having ten times more negative examples than positive ones.

For each fine-tuning task, we train all tested models for ten epochs, with negative entries changing every epoch during the training in the standard setting (see Section C.4.2 for an exception), as done for the pretraining tasks on this data. After the ten epochs, the best model is selected based on the validation F1-score, which is computed after every epoch. To enhance the robustness and reliability of our results, we train the models five times with different random seeds and average the values to obtain more stable results. All random decisions between runs are made deterministic, e.g., the same partition of the identities is applied within every training run. Similarly to the pretraining, we create a new IR-head, identical to the classical NSP-head for all fine-tuning tasks (i.e., we do not re-use the NSP/NMF/MFR heads learned during pretraining). We publish our evaluation code[8].

### C.3 Metrics

We report several standard classification and ranking metrics (Manning, 2009; Radlinski & Craswell, 2010).

- **Accuracy**: Proportion of correctly retrieved formulas among all retrievals.

- **Precision**: Proportion of retrieved formulas that are relevant.

- **Recall**: Proportion of relevant formulas that are retrieved.

- **$F_1$-Score**: Harmonic mean of precision and recall: $2 \cdot \dfrac{\text{precision} \cdot \text{recall}}{\text{precision} + \text{recall}}$.

---

[8]https://github.com/aieng-lab/transformer-math-evaluation

- **Precicion at $k$ (p@$k$)**: Precision among the top $k$ retrieved documents

- **Average Precision (AP)**: Average of p@$k$ values for all retrieved relevant documents, i.e.

$$AP = \frac{\sum_{k=1}^{n} p@k \cdot \text{relevant}(k)}{\sum_{k=1}^{n} \text{relevant}(k)}.$$

  Here, $n$ denotes the number of retrieved documents and relevant($k$) is 1 if the $k$-th ranked document is relevant, otherwise 0. The denominator is equal to the total number of relevant documents.

- **Normalized Discounted Cumulative Gain (nDCG)**: A score comparing the ranked list to an ideal ranking (Järvelin & Kekäläinen, 2002). It builds on Discounted Cumulative Gain (DCG), which measures how well relevant documents are ranked by giving higher scores to relevant documents that appear earlier in the list. The formula for DCG is (Burges et al., 2005):

$$DCG = \sum_{k=1}^{n} \frac{\text{relevance}(k)}{\log_2(i+1)},$$

  where relevance($k$) is the graded relevance of the document at position $k$. For this work, we use a binary relevance score. nDCG is the normalized DCG value computed by dividing through the maximum possible DCG (i.e., the ideal ranking).

For NMF, we use all unique names as queries, and for MFR, we randomly pick one true formula per mathematical identity to serve as the query for all paired formulas of that identity in the test set. In both cases, we compute the ranking metrics over all formulas associated with the same name or identity (including the challenging false examples). The final score is obtained by averaging the metrics across these evaluations. For binary metrics (accuracy, precision, recall, F1), we consider any predicted probability above 50% to indicate positive classification.

## C.4 NMF Retrieval Task

We present additional results based on the NMF fine-tuning task discussed in Section 7. The ability to conduct such fine-grained analysis highlights the strength of MAMUT, which tracks all applied transformations, allowing meta data to be used for analyzing their impact. We release the NMF-FT dataset with this additional meta data[9].

### C.4.1 Fine-Grained NMF Evaluation

We perform a fine-grained performance analysis of NMF-FT based on three criteria: the number of falsifying strategies applied (Table 18), whether symbol substitutions were applied (Table 19), and whether the input was given as a mathematical formula or as a textual description (Table 20).

Table 18 reveals how model accuracy varies with the number of falsifying strategies used for the test samples. Strategies with zero strategies correspond to the positive labeled entries. As expected, applying more strategies generally makes falsified samples easier to detect, resulting in higher accuracy. This suggests that future applications of MAMUT could focus more on applying fewer falsifying strategies to increase the task difficulty.

Table 19 breaks down accuracy by whether symbol substitutions were applied, divided into variables, functions, or both. As expected, applying substitutions slightly reduces accuracy, suggesting that these changes introduce a moderate challenge. Function substitutions appear to be slightly more challenging than variable substitutions. However, the smaller number of training samples with function substitution could also contribute to this discrepancy.

Table 20 highlights the performance gap between models when processing mathematical formulas versus their textual descriptions. In genera, models perform better when provided with textual descriptions. Importantly,

---

[9]https://huggingface.co/datasets/ddrg/named_math_formulas_ft

| Number of Strategies | Accuracy | Number of Test Samples |
|---:|:---:|---:|
| 0 | 99.39 | 2,626 |
| 1 | 90.01 | 2,626 |
| 2 | 97.44 | 8,352 |
| 3 | 99.23 | 7,075 |
| 4 | 99.79 | 3,176 |
| > 5 | 99.98 | 785 |

Table 18: Accuracy of NMF-FT across varying number of applied falsifying strategies. Results are averaged across all models (Table 6). Accuracy scores are reported as percentages. Note that 0 applied strategies corresponds to positive examples.

| Variant | Accuracy | | Number of Test Samples | |
|---|:---:|:---:|---:|---:|
| | ✓ | ✗ | ✓ | ✗ |
| Variable Substitution | 95.61 | 97.16 | 10,280 | 18,616 |
| Function Substitution | 94.43 | 96.71 | 1,279 | 27,617 |
| Symbol Substitution | 95.60 | 97.22 | 10,802 | 18,094 |

Table 19: Impact of symbol substitutions on NMF-FT accuracy. Results are averaged across all models (Table 6). Substitution types are variable, function, and both (symbol). Accuracy is reported seprarately for cases with (✓) and without (✗) substitution. Accuracies are reported as percentages.

| Model | Accuracy | | Difference |
|---|:---:|:---:|:---:|
| | Text | Formula | |
| BERT | 91.23 | 89.41 | 1.81 |
| MathBERT | 93.98 | 89.41 | 1.15 |
| MPBERT | 95.79 | 94.81 | 0.98 |
| MAMUT-BERT-MLM | 97.11 | 96.57 | 0.54 |
| MAMUT-MathBERT-MLM | 96.32 | 95.98 | 0.34 |
| MAMUT-MPBERT-MLM | 97.62 | 97.13 | 0.49 |
| MAMUT-BERT | 99.76 | 99.68 | 0.08 |
| MAMUT-MathBERT | 99.78 | 99.73 | 0.05 |
| MAMUT-MPBERT | 99.84 | 99.80 | 0.04 |

Table 20: Accuracy of NMF-FT models when evaluated on textual ($N = 18,764$) versus formula ($N = 10,132$) inputs. Results are averaged across all models (Table 6). All accuracy scores are reported as percentages.

| Model | Precision | Recall | F1 | p@1 | p@10 | AP | nDCG |
|---|---|---|---|---|---|---|---|
| MPBERT | **66.8** | 99.5 | **79.9** | **83.7** | **74.2** | **84.1** | **91.9** |
| MPBERT-random-falses | 14.0 | **99.7** | 24.6 | 17.1 | 16.3 | 19.0 | 53.7 |
| MPBERT-constant-falses | 59.5 | 99.2 | 74.4 | 76.2 | 70.3 | 78.9 | 89.3 |

Table 21: Results for the NMF fine-tuning task for different fine-tuning settings for MPBERT. All scores are reported as percentages.

our models pretrained on MAMUT-enhanced data exhibit a smaller gap between text and formula inputs, reflecting their pretraining on both, textual (MT) and formula (MF) MLM tasks.

We also considered analyzing accuracy by applied falsifying strategy types. However, we observed that strategies with more training samples tend to show higher accuracies, making it difficult to identify which strategies are more challenging.

### C.4.2 Ablation Study on Challenging False Examples

To assess the specific contributions of MAMUT in falsifying formulas, we also evaluate two modified fine-tuning versions of MPBERT. The standard setup involves challenging false examples generated by MAMUT, dynamically changing with each epoch. To assess the impact of MAMUT's falsification, we evaluate two controlled variants of MPBERT: one using randomly sampled (i.e., non-challenging) false examples from other identities that change each epoch (MPBERT-random-falses), and another using a fixed set of challenging false examples across all epochs (MPBERT-constant-falses). Note that the validation and test sets remain unchanged. Results are reported in Table 21, alongside the baseline MPBERT. The table confirms our intuition: both the difficulty of false examples and the dynamic sampling strategy contribute positively to model performance. While MPBERT-random-falses achieves the highest recall, likely due to retrieving nearly all documents, it performs very poorly on all other metrics, emphasizing the importance of challenging negatives during training.

### C.4.3 Mathematical Identity Analysis

Up to this point, we have only considered metrics across all mathematical identities in NMF. Now, we will discuss how the models perform on formulas based on a specific mathematical identity in NMF. Figure 8 shows the F1-score of NMF-FT for various models dependent on the original mathematical identity. In this figure, the identities are sorted by the average F1-score calculated from all shown models. Consequently, the most challenging identities are located on the left-hand side, while the easier ones are on the right. Notably, formulas containing many substitutable symbols are among the most challenging identities, such as Derivative of a Function $(f'(x) = \lim_{h \to 0} \frac{f(x+h)-f(x)}{h})$, Complex Number Sum $((a+bi) + (c+di) = (a+c) + (b+d)i)$, Determinant of 3x3 Matrix $(\det \begin{pmatrix} a & b & c \\ d & e & f \\ g & h & j \end{pmatrix} = a \cdot \det \begin{pmatrix} e & f \\ h & j \end{pmatrix} - b \cdot \det \begin{pmatrix} d & f \\ g & j \end{pmatrix} + c \cdot \det \begin{pmatrix} d & e \\ g & h \end{pmatrix})$, and Rule de l'Hôpital $(\lim_{x \to a} \frac{f(x)}{g(x)} = \lim_{x \to a} \frac{f'(x)}{g'(x)})$, while formulas with few substitutable symbols are among the easiest identities, such as Alternating Harmonic Series $(1 - \frac{1}{2} + \frac{1}{3} - \frac{1}{4} \pm \cdots = \ln(2))$, Covariance $(\mathrm{Cov}[X,Y] = \mathrm{E}[(X - \mathrm{E}[X])(Y - \mathrm{E}[Y])])$, De Morgan Law $(\forall x, y : \neg(x \wedge y) = \neg x \vee \neg y)$, and Eulers Formula $(\forall \alpha \in \mathbb{C} : e^{i\alpha} = \cos(\alpha) + i\sin(\alpha))$. Inequalities are rather challenging, while addition theorems appear to be easier.

When comparing the results of different models, we observe that the best models are able to classify most identities almost perfectly but do struggle for certain identities (e.g., Binomial Formulas), resulting in drops up to several percentage points in F1-score. In general, each model appears to have its own challenging identities, since the F1-scores of a particular model do not form a monotonic line across the mathematical identities in Figure 8.

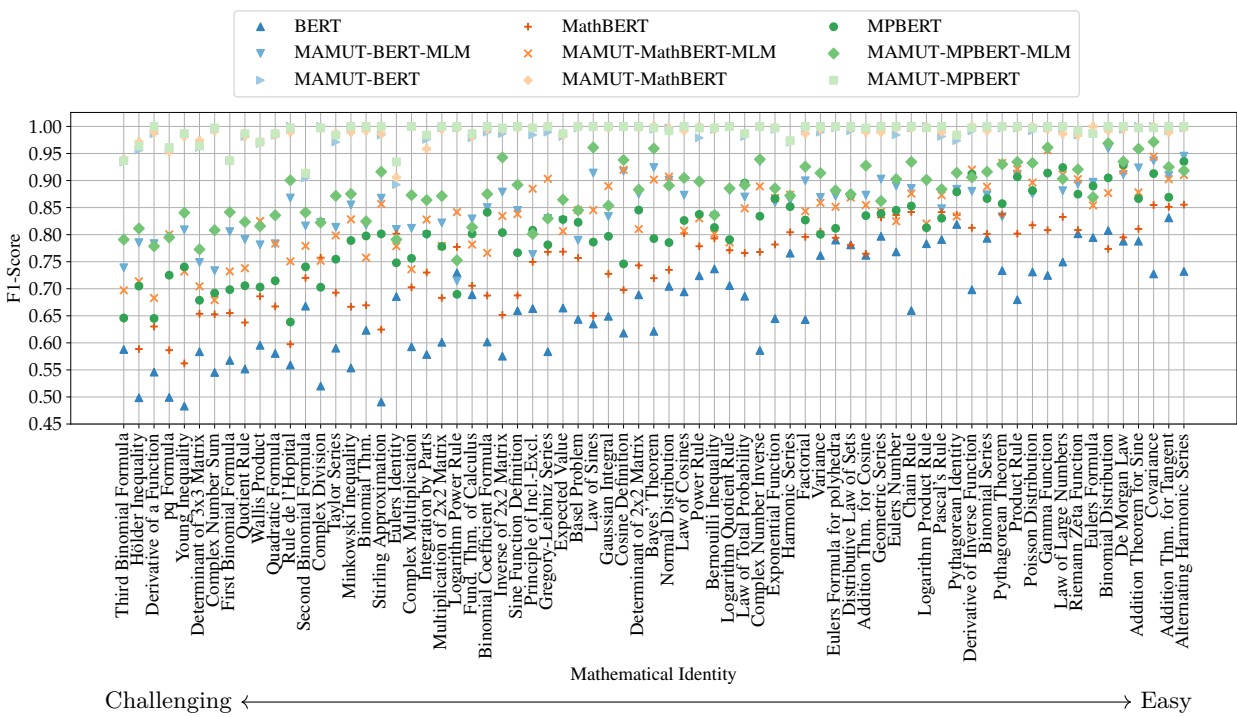

Figure 8: F1-Score of NMF-FT for various models split up by mathematical identities. The identities are sorted by the average F1-score across all models. to be updated with final models

| Model | Precision | Recall | F1 | p@1 | p@10 | AP | nDCG |
|---|---|---|---|---|---|---|---|
| BERT | 7.4 | 28.1 | 6.7 | 4.3 | 8.2 | 12.3 | 45.2 |
| MathBERT | 9.4 | 28.2 | 12.9 | 4.6 | 6.7 | 11.1 | 44.0 |
| MPBERT | 9.4 | 31.8 | 12.6 | 2.0 | 7.9 | 12.0 | 45.2 |
| MAMUT-BERT-MLM | 13.0 | 45.0 | 19.8 | 9.5 | 14.6 | 17.7 | 50.6 |
| MAMUT-MathBERT-MLM | 9.1 | 31.4 | 10.1 | 0.6 | 4.5 | 10.2 | 42.5 |
| MAMUT-MPBERT-MLM | **13.2** | **50.3** | **20.7** | **12.9** | **15.9** | **20.5** | **52.6** |
| MAMUT-BERT | 33.7 | 94.5 | 49.6 | 74.8 | 66.7 | 75.0 | 88.0 |
| MAMUT-MathBERT | 14.6 | 56.6 | 23.0 | 26.5 | 20.6 | 23.7 | 55.4 |
| MAMUT-MPBERT | 37.7 | 99.5 | 54.6 | 81.4 | 71.6 | 80.1 | 91.2 |

Table 22: Results for NMF-Split. All scores are reported as percentages. Since the models MAMUT-BERT, MAMUT-MathBERT and MAMUT-MathBERT include the training data of the NMF and MFR tasks in their pretraining, they have a strong advantage and are therefore highlighted in gray. However, we note that these models did not see the test data of NMF or MFR as well, i.e., there is no information leak leading to this performance. Still, we focus our analysis of the comparison of fine-tuned models that only used MLM as pretraining task to separate the impact of the data from that of the pretraining tasks. **Bold** highlights the best result per metric excluding the fully pretrained models.

### C.4.4 NMF-Split Fine-Tuning

We evaluate the models on a modified NMF task, denoted as NMF-Split. In this task, we further reduce the NMF-FT data, such that each mathematical identity appears in exactly one of the data splits. For example, all versions of the Pythagorean Theorem are included in the training data but not in its validation and test data. This setup allows us to assess the mathematical knowledge of the model. During fine-tuning, the model is only instructed on which type of objective to learn, and it must combine this learned objective with knowledge gained during pretraining for evaluation. For each of the five runs used for averaging the results, we apply a different partitioning of mathematical identities, resulting in 49 identities remaining in the training data, and eleven identities in both the test and validation set. During fine-tuning, we freeze the model parameters of BERT such that only the classification head is learned and we evaluate on actually learned core model knowledge.

The results are shown in Table 22. This task is clearly much more challenging than NMF-FT. Since the test set contains ten times more negative positive examples, the BERT model even performs worse than a naive *always true* classifier in terms of precision. Notably, the p@1 scores are consistently low across all models. However, MAMUT-variants of BERT and MPBERT demonstrate a clear improvement across all metrics. This suggests that these models *remember* mathematical identities better and are more effective at applying this knowledge to previously unseen formulas.

