# OpenReview forum: "MAMUT: A Novel Framework for Modifying Mathematical Formulas for the Generation of Specialized Datasets for Language Model Training"
_TMLR — Accepted by TMLR_

### Review · Reviewer_CfKK · 2025-04-06

**Summary Of Contributions:**

his paper introduces Math Mutator (MAMUT), a novel framework designed to generate diverse, high-quality datasets of mathematical formulas and texts by leveraging symbolic representations, substitutions, and falsifications of mathematical expressions in LaTeX. MAMUT addresses the challenges of limited diversity and scalability in existing mathematical datasets, which are crucial for training mathematical language models. Two algorithms, EquVG (Equivalent Version Generation) and FalseVG (Falsified Version Generation), for generating mathematically valid and invalid variations of formulas. Methods for parsing and manipulating LaTeX formulas into symbolic expressions using SymPy, enabling substitutions, falsifications, and randomized LaTeX printing.

**Audience:**

Yes

**Claims And Evidence:**

Yes

**Requested Changes:**

- Include experiments demonstrating how the generated datasets improve the performance of mathematical language models on tasks like MIR, equation completion, or formula embedding.

- Compare the diversity and quality of MAMUT-generated datasets with existing datasets, such as ARQMath, MATH, or GSM8K.

- Include a case study that fine-tunes an existing mathematical language model (e.g., MathBERT, LLMs like Qwen2 or LLaMA) on the MAMUT datasets and evaluates its performance on a relevant task.

- Provide an error analysis of the generated datasets, especially for falsified examples, to assess the quality and reliability of FalseVG.

**Strengths And Weaknesses:**

Strengths:

-The framework addresses a significant gap in mathematical language processing by automating the generation of diverse and challenging datasets.

-The use of symbolic expressions (via SymPy) to manipulate formulas ensures mathematical correctness and scalability.

-The datasets are well-documented, extensive, and target various tasks, making them highly useful for advancing mathematical language models.

Weaknesses:

-While the paper provides a statistical analysis of the datasets, there is no empirical evaluation of how these datasets improve the performance of mathematical language models.

-The paper lacks benchmarks or comparisons with existing mathematical datasets (e.g., MATH, ARQMath).

-The datasets are primarily tailored for LaTeX-based mathematical corpora. It is unclear how well MAMUT could generalize to other representations of mathematical notation (e.g., MathML).

-The paper does not provide an error analysis of the generated data, particularly for falsified examples. An analysis of failure cases would help identify potential biases or shortcomings in the framework.

---

> ### Author Response · Authors · 2025-04-27
> **Response to Reviewer CfKK**
>
> Thanks for your review. We tried to address everything and respond to the requested changes below. To further help with checking our response, the supplemental material contains a PDF, in which all changes (except typo corrections) for this revision are marked in blue.
>
> **Requested Changes:**
>
> - **Experiments:** We added extensive experiments (MIR tasks) in the revised manuscript, see Section 7. The results show that models trained on MAMUT-enhanced data outperform existing mathematical models.
> - **Comparison with Existing Datasets:** We have included a new discussion section, where we compare the diversity and quality of MAMUT-generated datasets (see Section 8.1), highlighting the notational variance introduced by MAMUT-enhanced datasets.
> - **Case Study for Training Mathematical Models:** We applied both, pretraining and fine-tuning to two existing mathematical models as part of our added experiments, see Section 7. We show that models with and without prior mathematical pre-training benefit from additional MAMUT-based training. This suggests that MAMUT introduces complementary patterns not present in prior mathematical corpora used for training.
> - **Error Analysis:** We aggree that an error analysis of MAMUT and its generated datasets would provide valuable insights into their quality and reliability. While this is outside the current scope of the paper, we recognize its importance and plan to include such an analysis in future work. It is worth noting that our generation process is fully rule-based, and we have already incorporated safety measures to avoid generating falsifyed examples that are actually equivalent, e.g. by avoiding transformations like adding zero, multiplying by one, or applying strategy distribute to $\cos(1\cdot x)$ (since $\cos(1)\cdot \cos(x) = 1\cdot \cos(x)=\cos(x)=\cos(1\cdot x)$ is actually equivalent and not falsified). However, despite these precautions, we acknowledge that edge cases may remain. We add a brief discussion of this point to the paper (see Section 9; Limitations).
>
> **Weaknesses:**
> Most of the mentioned weaknesses have been addressed in the sections above.
>
> - **Extension to Other Formats:** While MAMUT currently supports only LaTeX, it can easily be extended to support other formats like MathML. This is possible thanks to existing SymPy parsers and printers in various formats. We address this limitation in the revised manuscript (Section 9) and describe how the framework can be adapted to support additional formats.

---

### Review · Reviewer_RUmQ · 2025-04-10

**Summary Of Contributions:**

1. Focused on a novel, practical yet underexplored task: generating equivalent variants and inequivalent hard negatives given a mathematical formula to better train models' encoding of mathematical content.
2. Proposed a structured framework, **MAMUT**, to address the aforementioned task: parsing a LaTeX formula into a symbolic expression, mutating it, and outputting its LaTeX version. Mutations include:
	1. *EquVG*: Generating mathematically equivalent variants by symbol substitution based on predefined symbol groups;
	2. *Randomized LaTeX Printing*: Generating mathematically equivalent variants by randomizing LaTeX code based on predefined rules;
	3. *FalseVG*: Generating similar-looking but inequivalent variants by predefined modification strategies.
3. Manually curated NMFT dataset (composed of 71 named mathematical formulas) and subsequently constructed four larger-scale datasets, MF, MT, NMF, and MFR, for model training.

**Audience:**

Yes

**Broader Impact Concerns:**

None.

**Claims And Evidence:**

Yes

**Requested Changes:**

1. Typos.
	- Sec. 5, Redundant language "slightly modifies formulas slightly to"
	- Table 5. Inconsistency between "Generated Versions" in title and "Generated Entries" in caption.
2. (Critical for recommendation) **Add supporting experiments**, such as MIR comparison between a model trained using the constructed datasets and relevant baselines.
3. Add more details about SymPy implementation to the main text, especially verifying the validity and falsity of generated expressions.
4. (Additional comment, not a requested change) It seems that "symbol substitution" in EquVG bears a resemblance to $\alpha$-conversion in Definitional equality[1]. Hopefully, the task of "generating equivalent/falsified mathematical formula" could benefit from a formal verification perspective.

[1] https://rocq-prover.org/doc/v8.18/refman/proofs/writing-proofs/equality.html

**Strengths And Weaknesses:**

### Strengths
1. This work targets a practical yet underexplored task. This could enhance multiple tasks, including retrieval, AI4SCI, reasoning, etc. Identifying such a niche area that can have broad implications is commendable and shows the potential value of the research.
2. The proposed framework is technically sound: Parsing a LaTeX formula into an operator tree, mutating the operator tree, optionally verifying its validity by SymPy, and finally back-translating it into LaTeX (w/ equivalence-preserving randomization). This technical rigor indicates a well-thought-out approach to the problem at hand.
3. The curated datasets are both large in size and broad in domain (manually constructed + adapted from existing large-scale math datasets).

### Weaknesses
1. **Predefined mutation rules are relatively limited**:
	1. EquVG is limited to symbol substitution. However, in real-world scenarios, many more potential equivalent transformations could be considered, e.g., $1 + 2 + \dots + n$ and $\sum_{i=1}^n i$ (the elimination of ellipsis);
	2. Rules of LaTeX printing randomization are also limited, e.g., recursive application of the same function $f$ can be both denoted as $f^{(3)}(x)$ and $f(f(f(x)))$.

2. **Potential validity issues**. This work doesn't lay adequate emphasis on ensuring the validity of formulas generated by EquVG. The authors claim "the (general) validity of each used expression is verified using SymPy to ensure high data quality" in Sec. 6. However, SymPy check is insufficient for more complex scenarios. For example, variables of different domains/types might result in different behaviors:  $\lim_{n \rightarrow \infty} \sin(n\pi) = 0$ (human tend to assume $n \in \mathbb N$). Upon replacing $n$ to $x$, $\lim_{x \rightarrow \infty} \sin(x\pi)$ does not exist if $x\in \mathbb R$.

3. **Potential false negatives**. It appears that only the MF dataset, which is augmented by EquVG, has been verified using SymPy. The results of FalseVG have not been verified, which may lead to false negatives in the dataset.

4. **Insufficient baselines**. This submission claims that the MAMUT framework can augment mathematical data and subsequently enhance mathematical embedding. However, there is a lack of supporting experiments for this claim.

---

> ### Author Response · Authors · 2025-04-27
> **Response to Reviewer RUmQ**
>
> Thanks for your review. We tried to address everything and respond to the requested changes below. To further help with checking our response, the supplemental material contains a PDF, in which all changes (except typo corrections) for this revision are marked in blue.
>
> **Requested Changes:**
> 1. We fixed the typos.
> 2. We added exhaustive MIR experiments in the revised manuscript, see Section 7 Experiments. Based on these tasks, models trained on MAMUT-enhanced data outperform existing mathematical models, and even results in a notable increase in performance when an already mathematical model is further pre-trained, suggesting that MAMUT-enhanced data consists of novel properties compared to those used for training these mathematical models.
> 3. We add clarifying comments regarding the validity assurance of input formulas (see Section 6). Additionally, we add a discussion on the quality assurance of generated data (see Section 8.2).
> The desciption of the sympy implementation is very detailed and lengthy. Hence, we decided to keep most of this technical description stays in the Appendix. We further updated this description with a description about the usage of sympys solvers to ensure general validity (see Appendix B.1).
> 4. We thank the reviewer for pointing out the analogy of $\alpha$-conversions to MAMUT's symbol substitutions. We now briefly mention this connection in Section 5.1 of the revised manuscript. Furthermore, one could interpret the falsifying strategy Variable as a form of *inconsistent* $\alpha$-conversion$. However, we believe that explicitly including this interpretaion in the paper may be more confusing than helpful for the intended audience.
>
> **Weaknesses:**
> 1. EquVG is not limited to symbol substitutions as it involves also randomized printing. As seen in Table 5, more than 38 versions have been generated on average per base formula, indicating diverse generation options. While we acknowledge that our implementation is not an exhaustive coverage of all possible equivalent notations (e.g., ellipsis handling, recursive function notation), this limitation is inherent to any such system. To address this, our framework is modular and extensible: we defined already over 60 printing variations (see sympy-random-LaTeX/sympy/settings.py), and new variants can be easily incorporated via custom extensions to `LatexPrinter` (sympy-random-LaTeX/sympy/printing/latex.py). We now explicitly acknowledge this limitation and extensibility in the new Limitations section (Section 9).
> 2. We indeed use sympys validity checks to ensure the validity of input formulas when constructing the MF dataset. Specifically, formulas are only used if `sympy.simplify()` returns `True` (indicating general validity) or if an implication can be created based on a general solution found with `sympy.solve()`. The example mentioned by the reviewer would not pass this test as it is not generally valid. However, for NMF and MFR, formulas are created manually and include metadata specifying which variables may be substituted. These datasets are intended for a broad retrieval task, where any formula that could match a certain formula should be retrieved (e.g., $x!=1\cdot 2\cdot \dots x$ should be retrieved for query *Factorial* despite the unusual notation).
> 3. You are correct that we do not verify that falsified formulas are non-equivalent to the original formula. While we implemented some careful safeguards to avoid incorrect falsification (e.g., avoiding transformations such as adding zero, multiplying by one, or applying strategy distribute to $\cos(1\cdot x)$ as $\cos(1)\cdot \cos(x) = 1\cdot \cos(x)=\cos(x)=\cos(1\cdot x)$ is actually equivalent and not falsified), we acknowledge that some rare edge cases may still exist. We now point this out in the new Limitations section (see Section 9).
> 4. We now added supporting experiments with mathematical baselines (see Section 7).

---

> > ### Comment · Reviewer_RUmQ · 2025-04-30
> >
> > Thank you authors for your thorough and comprehensive response! I appreciate the effort you've put into addressing the concerns.
> >
> > To provide further clarity,
> > - Weakness-1 is not intended to criticize the current work but a potential direction for future exploration;
> > - Requested Change-4 is merely an additional comment suggesting a possible future direction, rather than an actual requested change. I agree that explicitly including $\alpha$-conversion might confuse the target audience.
> >
> > All my concerns are satisfactorily resolved in this detailed response and the subsequent revision. I'll update my "Claims And Evidence" to "Yes" and am inclined to support the acceptance of this work :)

---

### Review · Reviewer_krLM · 2025-04-16

**Summary Of Contributions:**

The author proposes the first systematic mathematical formula mutation framework, MAMUT, which supports symbol substitution, syntactic rewriting, and error injection, significantly enhancing data diversity. Additionally, based on this framework, four large-scale datasets are proposed to help improve the model's ability to understand and represent mathematical formulas.

**Audience:**

Yes

**Broader Impact Concerns:**

The work does not raise concerns regarding ethical implications. I believe this is a meaningful contribution to the community.

**Claims And Evidence:**

Yes

**Requested Changes:**

1. Increase the number of model comparison experiments on the dataset. This is a change that I particularly value.
2. **Recommend adding more discussions on the quality assessment of the dataset**. Meaningful discussions will be key to recommending acceptance.
3. The manuscript is an interesting piece of work. The authors have put in a substantial amount of effort in creating the dataset. However, more comparisons with other existing mathematical benchmarks (such as MathVista) are needed to enhance the expressiveness of the work.
4. The figures could incorporate more visual elements. This will enhance the readers' understanding.

**Strengths And Weaknesses:**

**Strengths:**
+ S1: Aiming at the limitations of existing datasets in terms of scalability and diversity, the author generates different LaTeX representations by randomly changing variable and function identifiers and utilizing mathematical properties. This is an interesting piece of work.
+ S2: The author has open-sourced most of the code, and the appendix materials are quite comprehensive.
+ S3: The proposed method is clear, and easy-to-understand examples are provided at the same time.

**Weaknesses:**
+ W1: It seems that the performance of existing models (such as MathBERT) on the proposed BMK has not been given.
+ W2: Regarding how to evaluate the quality of the generated datasets, I think the proposed method is similar to using existing large language models for rollout and then conducting rule matching and extraction. High-quality answers are the key to improving my evaluation score.
+ W3: Equivalent version generation and error version generation are used to enhance the performance of the dataset. However, these operations seem to be exhaustible? But in reality, mathematical operations are difficult to exhaust.
+ W4: How can we ensure that when generating equivalent operations, the model's understanding of mathematical formulas is based on deep semantics rather than shallow similarity features?

---

> ### Author Response · Authors · 2025-04-27
> **Response to Reviewer krLM**
>
> Thanks for your review. We tried to address everything and respond to the requested changes below. To further help with checking our response, the supplemental material contains a PDF, in which all changes (except typo corrections) for this revision are marked in blue.
>
> **Requested Changes:**
>
> 1. **Model Comparisons** We have added experiments comparing models trained on our MAMUT-enhanced datasets with existing mathematical models (see Section 7, *Experiments*). These experiments show that models trained on our data outperform existing baselines. Further, the results show that further pre-training with our data also improves the existing baselines.
> 2. **Quality Assessment Discussion:** We added a discussion on data quality in Section 8.2, highlighting our efforts to maximize data quality, while pointing out potential threats (see also Section 9, *Limitations*).
> 3. **Comparison to Existing Datasets:** We now include a discussion of existing mathematical datasets in Section 8.1, highlighting the notational variance of MAMUT. While we did not explicitely mention MathVista due to its focus on visual problem-solving, which targets a different modality, we briefly discuss the difference of MAMUT-generated datasets with textual mathematical datasets, such as GSM8K, MATH, and AMPS.
> 4. **Improved Visual Presentation:** We revised the figures to include more visual elements, such as icons and symbol highlights, to improve clarity and aid comprehension.
>
>
> **Weaknesses:**
>
> - **W1:** While our primary goal is *not* to propose a new benchmark (BMK) but to provide data for improving mathematical representations in LMs, we have now included experiments (see Section 7, *Experiments*) supporting our claims.
> - **W2:** We respectfully disagree that MAMUT is similar to LLM-based augmentation methods. While these LLM pipelines typically generate paraphrased or slightly changed problems (e.g., using different numbers), MAMUT enriches the notational variance of formulas, keeping the text parts constant. MAMUT offers advantages over LLM-based methods such as fine-grained control (e.g., applying certain transformation or randomized LaTeX printing rules) and transparency (by tracking applied transformations as meta data).
> - **W3:** We agree that rule-based transformations are inherently finite. However, our framework is the first to systematically study such symbolic equivalent and falsification strategies. Despite their bounded nature, our rules already span a wide variety of common transformations and errors. Nevertheless, we mention this in the new limitation section (see Section 9).
> - **W4:** MAMUT is explicitly designed to minimize shallow feature reliance. For example, in the MFR-based MIR task (query and documents are formulas), query and formulas typically don't share the same variable names and use different notation variants to minimize lexical overlap. This enforces the model to focus on underlying semantic equivalence. While we cannot guarantee deep understanding, our dataset design along with previous findings (e.g., [1] showed that variable renaming in query and documents for MIR helps to make the model's decision rely more on the variables) suggests that MAMUT increases the likelihood of learning true semantics over shallow features.
>
>
> [1] Reusch, A., Gonsior, J., Hartmann, C., & Lehner, W. (2024, March). Investigating the usage of formulae in mathematical answer retrieval. In European Conference on Information Retrieval (pp. 247-261). Cham: Springer Nature Switzerland.

---

> > ### Comment · Reviewer_krLM · 2025-05-18
> >
> > + The author has addressed most of my concerns, and I also respect the significant contributions they have made to the community. However, based on the comments from other reviewers, I would like to see the author provide results on tasks using Llama or Qwen2 to verify the claim that “MAMUT outperforms LLMs.”
> > + Overall, I am inclined to accept this work, and I hope the author can provide further supplements.

---

> > > ### Comment · Action_Editor_HcPY · 2025-06-01
> > >
> > > Dear Reviewer krLM,
> > >
> > > Can you help to upload your official recommendation with the recommendation button so that we can move forward the review process?
> > >
> > > Best,
> > > AE

---

### Review · Reviewer_AzAm · 2025-04-24

**Summary Of Contributions:**

This work aims to enhance quantity and quality of mathematical fomulas. It proposes a framework to modify existing math formulas considering several property and keys of this kind of data. Based on the proposed framework, quantity of math formulas are increased for hundreds times.

**Audience:**

Yes

**Claims And Evidence:**

Yes

**Requested Changes:**

Authors are expected to present experiments to show effectiveness of proposed data framework in training LLMs such ChatGPT, Claude, Qwen, InternLM, and etc.
Besides the normal LLMs for plain texts, LLMs for code that can process LaTex also should be included in comparison.

**Strengths And Weaknesses:**

Strengths:
1. Focusing on a special kind of text data, math formula, and analyzed its key difference from raw texts. This kind of data is quite important in reasoning and scientific domains.
2. By proposing rule-based framework in detail, the proposed framework MAMUT can cover a large protion of math formulas, so that it can generate data of enough quantity.

Weakness:
1. The target of this work is to increase diversity of math formula and further use it as text to help training LLMs. But weakness of LLM in processing math formulas can somehow solved by processing codes, such as Latex. Therefore, the necessary of this work is kind of questionable.
2. This work focuses on mathematical formulas, while formulas are also common in fields like physics. Current formula corpus seems to be weak as they are from a single field.
3. Author provides quantities and some statistical results about the generated dataset, but its effectiveness is not examined by experiments, making this work incomplete.

---

> ### Author Response · Authors · 2025-04-27
> **Response to Reviewer AzAm**
>
> Thanks for your review. We tried to address everything and respond to the requested changes below. To further help with checking our response, the supplemental material contains a PDF, in which all changes (except typo corrections) for this revision are marked in blue.
>
> **Requested Changes:**
>
> While training large LLMs is an interesting direction for future work, we have now added a new experiment section based on BERT models in the revised manuscript (see Section 7, *Experiments*). We focus on BERT, specifically models based on bert-base-cased, for several reasons:
> 1. **Computational Efficiency:** They require significantly fewer computational resources compared to LLMs.
> 2. **Strong baselines:** Strong mathematical baselines [1, 2] based on bert-based-cased exist, allowing for direct comparisons.
> 3. **Precise evaluation:** BERT enables more controlled and precise evaluation (e.g., classification and ranking) and precision is crucial in mathematical tasks. Mathematical models, such as DeepSeekMath [3] and MathCoder [4], are evaluated on string equality of the final answer in benchmarks like MATH [5] and GSM8K [6]. The limitation of this approach is that it does not consider mathematical equivalence. For example, one solution to one of the provided MATH example problems is $\frac{1-\sqrt{2}}{2}$, but this expression is also commonly written as $\frac{1}{2}-\frac{1}{\sqrt{2}}$. Letting models deal with the recoginition of such equivalent formulations is the key purpose of the MAMUT data, which is why we believe that a retrieval based evaluation is more appropriate.
> 4. **Practicality of simpler encoder models:** While LLMs show impressive general-purpose capabilities, simpler models like BERT remain highly relevant due to their computational efficiency [9]. For example, they can be used to build a mathematical information retrieval system with enhanced mathematical understanding from our pretrained models.
>
> Our new experiments show that pretraining on MAMUT-enhanced data improves performance on mathematical tasks, supporting our main claim. Scaling to larger LLMs is an exciting future direction. However, pre-training LLMs with this data in addition goes beyond the the scope of this paper, which primarily aims to introduce the data generation framework and demonstrate the utility of the resulting data for the training of mathematical models.
>
>
> **Weaknesses:**
>
> 1. While we agree that improved LaTeX parsing is helpful for the processing of mathematical formulas, we respectfully disagree with the suggestion that improving LaTeX parsing alone is sufficient. Mathematical understanding involves more than simply parsing symbols, which LaTeX parsing primarily addresses. Therefore, our work introduces various levels of variation: randomized LaTeX printing to enhance LaTex processing capabilities (as mentioned), symbol substitutions to teach the notion of symbols as placeholders, and equivalence-based transformations (e.g., using commutativity or symmetry). These strategies go beyond simple LaTeX formatting and are designed to improve a model's deep mathematical understanding.
> 2. We acknowledge that our work focuses on mathematical formulas, as do many studies in this field [5-8].
> Extending our framework to other fields, such as physics or chemistry, may be possible for certain components (e.g., randomized printing, symbol substitutions for free variables), but it requires domain-specific knowledge. For instance, in physics, certain symbols have fixed meaning like $c$ (speed of light) or $h$ (Planck constant). Handling such cases would require special treatment, similar to how we addressed Euler's constant $e$ in our approach. Thus, while the rule-based approach for equation re-writing from MAMUT can be extended to other domains, this would be non-trivial and is beyond the scope of this paper.
> 3. We have now added experimental results in the updated manuscript (see Section 7, *Experiments*), supporting our claims using BERT models and demonstrating the effectiveness of the MAMUT-enhanced data.
>
>
> [1] MathBERT, DOI: 10.48550/arXiv.2106.07340
>
> [2] MathPreTrainedBERT, DOI: URN: urn:nbn:de:0074-3180-7
>
> [3] DeepSeekMath: DOI: 10.48550/arXiv.2402.03300.
>
> [4] Mathcoder, DOI: 10.48550/arXiv.2310.03731.
>
> [5] MATH, DOI: 10.48550/arXiv.2103.03874.
>
> [6] DOI: 10.48550/arXiv.2110.14168.
>
> [7] ARQMath, DOI: 10.1145/3483382.3483388
>
> [8] HARDMATH, DOI: 10.48550/arXiv.2410.09988.
>
> [9] ModernBERT, DOI: 10.48550/arXiv.2412.13663.

---

> > ### Comment · Reviewer_AzAm · 2025-05-06
> >
> > Thanks for authors' response. As experiments with proposed datasets, the effectiveness of MAMUT is demonstrated well. Results of table 6 in updated manuscripts present good improvement of BERT-style models by training on MAMUT. However, evaluations are conducted on generated datasets, which may results in overfitting, is it possible to evaluate on some in-the-wild benchmarks? If so, in-the-wild results will be more convincible.
> >
> > In conclusion, updated manuscript by authors has resolved most of my concern and made this work complete and I will update my original review. I think in-the-wild evaluation will enhance this work further, and I hope authors can consider such exmepriments.

---

### Review · Reviewer_dURn · 2025-05-20

**Summary Of Contributions:**

This paper proposed MAMUT (Math Mutator): a framework developed to create diverse and specialized datasets for training language models on mathematical content. The core idea is to produce: 1. Mathematically Equivalent Variants (EquVG) – through symbolic transformations such as commutativity, associativity, variable/function substitution, and LaTeX formatting variants. 2. Falsified Variants (FalseVG) – formulas that look similar but are mathematically incorrect, used as hard negatives for classification or retrieval tasks.

**Audience:**

Yes

**Broader Impact Concerns:**

No concern

**Claims And Evidence:**

Yes

**Requested Changes:**

- benchmark improvement in large scale LLM performance.
- Quality evaluation especially human evaluation on the generated datasets

**Strengths And Weaknesses:**

**Strengths**:
- Fills a key gap in symbolic mathematical data for LLMs.
- Well-structured augmentation using symbolic parsing, semantic grouping, and diverse LaTeX printing.
- with >30 versions per formula on average. Supports multiple downstream tasks**: MLM, NSP, contrastive learning, MIR.
- *Code and data available.

**Weakness**
- Does not benchmark improvement in large scale LLM performance.
- Lack of quality evaluation especially human evaluation on the generated datasets
- Complex LaTeX formulas might fail to parse or generate valid variants.

---

> ### Author Response · Authors · 2025-05-22
> **Response to Reviewer dURn**
>
> Thanks for your review.
>
> **LLM Evaluation:**
> While training large LLMs is an interesting direction for future work, it goes beyond our available computational resources.
> Moreover, we argue that focusing on BERT, as done in our study, offers several concrete advantages:
>
> 1. **Computational Efficiency:** They require significantly fewer computational resources compared to LLMs.
> 2. **Strong baselines:** Strong mathematical baselines [1, 2] based on bert-based-cased exist, allowing for direct comparisons.
> 3. **Precise evaluation:** BERT enables more controlled and precise evaluation (e.g., classification and ranking) and precision is crucial in mathematical tasks. Encoder models like BERT are also particularly well-suited for ranking tasks, which is the natural evaluation setup for assessing the impact of FalseVG. In contrast, mathematical LLMs, such as DeepSeekMath [3] and MathCoder [4], are evaluated on string equality of the final answer in benchmarks like MATH [5] and GSM8K [6]. The limitation of this approach is that it does not consider mathematical equivalence. For example, one solution to one of the provided MATH example problems is $\frac{1-\sqrt{2}}{2}$, but this expression is also commonly written as $\frac{1}{2}-\frac{1}{\sqrt{2}}$. Letting models deal with the recognition of such equivalent formulations is the key purpose of the MAMUT data, which is why we believe that a retrieval-based evaluation is more appropriate.
> 4. **Practicality of simpler encoder models:** While LLMs show impressive general-purpose capabilities, simpler models like BERT remain highly relevant due to their computational efficiency [7]. For example, they can be used to build a mathematical information retrieval system with enhanced mathematical understanding from our pretrained models.
>
> Our experiments show that pretraining on MAMUT-enhanced data improves performance on mathematical tasks, supporting our main claim. While scaling to LLMs is an exciting future direction, pretraining such models lies outside the scope of this paper.
> Our primary goal is to introduce a novel data generation framework and demonstrate the utility of the resulting data in training mathematically enhanced models.
>
>
> **Quality Evaluation:**
>
> Human evaluation is an interesting direction for future work, particularly for assessing the labeling quality in NMF and MFR.
> However, as shown by the dataset examples in Tables 14 and 15, labeling can be non-trivial due to the challenging negative examples and the high variance in input formulas.
> For instance, due to the falsifying strategy variable, consistent use of variables must be carefully checked across the entire formula, and recognizing equivalence or non-equivalence often demands a solid understanding of mathematics, given the many syntactically different ways a formula can be expressed. As a result, manual labeling is not only time-consuming and error-prone, but also cognitively demanding.
>
> Rather than verifying each generated formula manually (or a subset), we focus on ensuring high data quality at the input level (e.g., through filtering) and the design of rule-based transformation strategies that focus on sound transformations. For example, we explicitly avoid transformations that could accidentally preserve equivalence in FalseVG, such as adding zero, multiplying by one, or applying strategy distribute to expressions such as $\cos(1\cdot x)$ (since $\cos(1)\cdot \cos(x) = 1\cdot \cos(x)=\cos(x)=\cos(1\cdot x)$ is actually equivalent and not falsified).
> These safeguards and other data quality considerations are discussed in Section 8.2.
>
> We also explicitly acknowledge the absence of human evaluation as a limitation in Section 9.
> Given the complexity of the labeling task, the extensive quality controls we apply, and the primary focus of this work, we argue that a comprehensive human study is beyond the scope of this paper.
> Our main goal is to introduce a theoretical and practical framework for generating diverse equivalent and falsified formula variants, and to establish the foundation for future work in this space.
>
>
> [1] MathBERT: DOI 10.48550/arXiv.2106.07340
>
> [2] Reusch, A., Thiele, M., & Lehner, W. (2022). Transformer-Encoder and Decoder Models for Questions on Math. In CLEF (Working Notes) (pp. 119-137).
>
> [3] Deepseekmath: DOI 10.48550/arXiv.2402.03300
>
> [4] MathCoder: DOI 10.48550/arXiv:2310.03731
>
> [5] MATH: DOI 10.48550/arXiv:2103.03874
>
> [6] GSM8K: DOI  10.48550/arXiv:2110.14168
>
> [7] ModernBERT: DOI 10.48550/arXiv:2412.13663

---

### Decision · Action_Editor_HcPY · 2025-06-18

**Recommendation:** Accept as is

**Audience:**

Yes

**Audience Explanation:**

This work addresses the deficiency in the dataset for LaTeX formulas. ML audience will be interested in math language model training.

**Claims And Evidence:**

Yes

**Claims Explanation:**

This paper introduces Math Mutator (MAMUT), a novel framework designed to generate diverse, high-quality datasets of mathematical formulas and texts by leveraging symbolic representations, substitutions, and falsifications of mathematical expressions in LaTeX. MAMUT addresses the challenges of limited diversity and scalability in existing mathematical datasets, which are crucial for training mathematical language models. Two algorithms, EquVG (Equivalent Version Generation) and FalseVG (Falsified Version Generation), for generating mathematically valid and invalid variations of formulas. Methods for parsing and manipulating LaTeX formulas into symbolic expressions using SymPy, enabling substitutions, falsifications, and randomized LaTeX printing.

After the rebuttal, three reviewers are positive about the paper while the remaining reviewer slightly disagrees, considering that the currently focused language models in this paper are mostly BERT-like ones.

The main claim (the MAMUT framework and the constructed datasets enhance math encoding) are supported by the experiments in Sec. 7 (especially MLM-pretrained experiments). The AE considers that the proposed claims are well supported the experiments.